# Unsupervised Reinforcement Learning by Maximizing Skill Density Deviation

## Abstract

Unsupervised Reinforcement Learning (RL) aims to discover diverse behaviors that can accelerate the learning of downstream tasks. Previous methods typically focus on entropy-based exploration or empowerment-driven skill learning. However, entropy-based exploration struggles in large-scale state spaces (e.g., images), and empowerment-based methods with Mutual Information (MI) estimations have limitations in state exploration. To address these challenges, we propose a novel skill discovery objective that maximizes the deviation of the state density of one skill from the explored regions of other skills, encouraging inter-skill state diversity similar to the initial MI objective. For state-density estimation, we construct a novel conditional autoencoder with soft modularization for different skill policies in high-dimensional space. To incentivize intra-skill exploration, we formulate an intrinsic reward based on the learned autoencoder that resembles count-based exploration in a compact latent space. Through extensive experiments in challenging state and image-based tasks, we find our method learns meaningful skills and achieves superior performance in various downstream tasks.

## 1 Introduction

Reinforcement Learning (RL) has achieved remarkable success in game AI (Silver et al., 2018; Ye et al., 2021), autonomous cars (Cao et al., 2023; Wu et al., 2022), and embodied agents (Hansen et al., 2022; Miki et al., 2022). Traditionally, RL agents rely on well-designed reward functions to learn specific tasks (Luo et al., 2023). However, designing these reward functions is resource-intensive and often requires domain-specific expertise (Kwon et al., 2023; Gu et al., 2023), making the learned policies dependent on handcrafted rewards and potentially unable to capture the complexity of real-world scenarios. This reliance limits the agent's generalization capability across diverse tasks and results in poor adaptability. In contrast, recent advances in Large Language Models (LLMs) (Han et al., 2021; Achiam et al., 2023) signify that unsupervised auto-regression has led to powerful pre-trained language models, which can be adapted to downstream tasks via supervised fine-tuning (Ouyang et al., 2022; Touvron et al., 2023). A powerful vision encoder can also be pre-trained via masked prediction without annotations or labels (He et al., 2022; Bardes et al., 2024; Grill et al., 2020), and the encoder can be used to solve various vision tasks (Majumdar et al., 2023; Nair et al., 2023). Inspired by these breakthroughs, it is desirable to further explore similar unsupervised learning methods within the RL field. The goal is for unsupervised RL to learn useful behaviors in the absence of external rewards, thus equipping them with the capacity to quickly adapt to new tasks with limited interactions (Laskin et al., 2021).

The formulation of unsupervised RL has been studied in many prior works, which can be roughly categorized into empowerment-based skill discovery (Gregor et al., 2016) and pure exploration methods (Liu & Abbeel, 2021b). Empowerment-based methods aim to maximize the Mutual Information (MI) between states and skills, and the MI term can be estimated by different variational estimators (Song & Ermon, 2020). These methods have shown effectiveness in learning discriminative skills for state-based locomotion tasks (Eysenbach et al., 2019). However, the learned skills often have limited state coverage due to the inherent sub-optimality in the MI objective (Yang et al., 2023), which can lead to sub-optimal adaptation performance in downstream tasks and becomes more severe in large-scale state space (Park et al., 2024). Recent works introduce additional techniques like Lipschitz constraints and metric-aware abstraction to enhance the exploration abilities (Park et al., 2022; 2023; 2024). Pure exploration methods encourage the agent to explore the envi-

ronment with maximum state coverage; however, this can lead to extremely dynamic skills rather than meaningful behaviors for downstream tasks (Liu & Abbeel, 2021b; Laskin et al., 2022). Meanwhile, both the MI estimator and entropy estimation are not directly scalable to large-scale spaces, such as pixel-based environments (Rajeswar et al., 2023; Park et al., 2024).

To overcome the aforementioned limitations, this work proposes a novel skill discovery method by maximizing the *State Density Deviation of Different skills* (**SD3**). Specifically, we construct a conditional autoencoder for state density estimation of different skills in high-dimensional state spaces. Each skill policy is then encouraged to explore regions that deviate significantly from the state density of other skills, which encourages *inter-skill diversity* and leads to discriminative skills. For a stable state-density estimation of significantly different skills, we adopt soft modularization for the conditional autoencoder to make the skill-conditional network a weighted combination of the shared modules according to a routing network determined by the skill. We show the skill-deviation objective of SD3 resembles the initial MI objective in a special case. Further, to incentivize *intra-skill exploration*, we formulate an intrinsic reward from the autoencoder based on the learned latent space, which extracts the skill-relevant information and is scalable to large-scale problems. Theoretically, such an intrinsic reward is closely related to the provably efficient count-based exploration in tabular cases. To summarize, SD3 encourages inter-skill diversity via density deviation and intra-skill exploration via count-based exploration in a unified framework. We conduct extensive experiments in Maze, state-based Unsupervised Reinforcement Benchmark (URLB), and challenging image-based URLB environments, showing that SD3 learns exploratory and diverse skills.

Our contribution can be summarized as follows. (i) We propose a novel skill discovery objective based on state density deviation of skills, providing a straightforward way to learn diverse skills with different state occupancy. (ii) We propose a novel conditional autoencoder with soft modularization to estimate the state density of significantly different skills stably. (iii) The learned latent space of the autoencoder provides an intrinsic reward to encourage intra-skill exploration that resembles count-based exploration in tabular MDPs. (iv) Our method achieves state-of-the-art performance in various downstream tasks in challenging URLB benchmarks and demonstrates scalability in image-based URLB tasks. The open-sourced code is available at `https://github.com/s7p77/SD3`.

## 2 Preliminaries

**Markov Decision Process**  A Markov Decision Process (MDP) constitutes a foundational model in decision-making scenarios. We consider the process of an agent interacting with the environment as an MDP with discrete skills, defined by a tuple $(\mathcal{S}, \mathcal{A}, \mathcal{Z}, \mathcal{P}, r, \gamma)$, where $\mathcal{S}$ is the state space, $\mathcal{A}$ is the action space, $\mathcal{Z}$ is the skill space, $\mathcal{P} : \mathcal{S} \times \mathcal{A} \to \Delta(\mathcal{S})$ is the transition function, $r : \mathcal{S} \times \mathcal{A} \to \mathbb{R}$ is the reward function, and $\gamma$ is the discount factor. In this work, we consider a discrete skill space $\mathcal{Z}$ that contains $n$ skills since calculating the skill density deviation requires density estimation of all skills, while SD3 can also be extended to a continuous skill space by sampling skills from a continuous distribution for approximation. In each timestep, an agent follows a skill-conditional policy $\pi(a|s, z)$ to interact with the environment. Given clear contexts, we refer to 'skill-conditional policy' as 'skill'.

**Unsupervised RL**  Unsupervised RL typically contains two stages: unsupervised pre-training and fast policy adaptation. In the unsupervised training stage, the agent interacts with the environment without any extrinsic reward. The policy $\pi(a|s, z)$ is learned to maximize some intrinsic rewards $r_t$ formulated by an estimation of the MI term or the state entropy. The aim of unsupervised pre-training is to learn a set of useful skills that potentially solve various downstream tasks via fast policy adaptation. In the adaptation stage, the policy $\pi(a|s, z^\star)$ with a chosen skill $z^\star$ is optimized by RL algorithms with certain extrinsic rewards to adapt to specific downstream tasks. In the following, we denote $I(\cdot; \cdot)$ by the MI between two random variables and $\mathcal{H}(\cdot)$ by either the Shannon entropy or differential entropy, depending on the context. We use uppercase letters for random variables and lowercase letters for their realizations. We denote $d^\pi(s) \triangleq (1 - \gamma) \sum_{t=0}^{\infty} \gamma^t P(s_t = s|\pi)$ as the normalized probability that a policy $\pi$ encounters state $s$.

The empowerment-based skill discovery algorithms try to estimate the MI between $S$ and $Z$ via $I(S; Z) = \mathbb{E}_{z \sim p(z), s \sim p^\pi(s|z)}[\log p(z|s) - \log p(z)]$. Given the computational challenges associated with the posterior $p(z|s)$, a learned skill discriminator $q_\phi(z|s)$ is employed (Eysen-

bach et al., 2019) and a variational lower bound is established for the MI term as $I(Z; S) \geq \mathbb{E}_{z \sim p(z), s \sim p^\pi(s|z)}[\log q_\phi(z|s) - \log p(z)]$. Alternatively, pure exploration methods estimate state entropy by summing the log-distances between each particle and its $k$-th nearest neighbor, as $\mathcal{H}(s) \propto \sum_{s_i} \ln \|s_i - \mathrm{NN}_k(s_i)\|$.

## 3 METHOD

In this section, we first introduce the proposed SD3 algorithm that performs skill discovery by maximizing inter-skill diversity via state density estimation. Next, we present the formulation of intrinsic rewards for intra-skill exploration. Finally, we provide a qualitative analysis of SD3.

### 3.1 SKILL DISCOVERY VIA DENSITY DEVIATION

We develop our skill discovery strategy from a straightforward intuition: The explored region of each skill should deviate from other skills as far as possible. Formally, the optimizing objective for skill discovery, denoted as $I_{\mathrm{SD3}}$ and referred to as *density deviation*, is defined by

$$I_{\mathrm{SD3}} \triangleq \mathbb{E}_{z \sim p(z), s \sim d_z^\pi(s)} \left[ \log \frac{\lambda \, d_z^\pi(s)}{\lambda \, d_z^\pi(s)p(z) + \sum_{z' \neq z} d_{z'}^\pi(s)p(z')} \right], \quad (1)$$

where $z$ is sampled from $p(z)$, $s$ is sampled from the state distribution induced by the skill policy $\pi(a|s, z)$, and $\lambda > 0$ is a weight parameter. The numerator $d_z^\pi(\cdot)$ is the state density of skill $z$, and the denominator is the weighted average of the state density of $z$ and those of other skills $\{z'\}$. Since we uniformly sample skills from the skill set that contains $n$ skills, we have $p(z) = 1/n$ for each skill $z$. According to Eq. (1), it is easy to check that $I_{\mathrm{SD3}}$ attains its maximum when $\sum_{z' \neq z} d_{z'}^\pi(s) \to 0$ for all $(s, z)$ such that $p(z) \cdot d_z^\pi(s) > 0$, and the maximum value is $\mathcal{H}(Z)$. In this case, the state $s \sim d_z^\pi(\cdot)$ visited by skill $z$ has *zero* visitation probability by other skills, which means the explored regions of all skills do not overlap, and the learned skills are fully distinguishable. However, enforcing such a strong objective to separate the overlapping explored areas of skills may lead to limited state coverage for each skill. In extreme cases, each skill might only visit a distinct state that other skills do not access. Although this leads to distinguishable skills, the overall state coverage becomes overly limited, making them undesirable for learning meaningful behaviors.

In SD3, we adopt two mechanisms for addressing this problem. (i) A weight parameter $\lambda$ is used in the learning objective to regularize the gradients of $I_{\mathrm{SD3}}$ to other skills. To see this, for each $(s, z)$, we denote the state density of other skills $\{z'\}$ except $z$ as $\rho_{z^c} \triangleq \sum_{z' \neq z} d_{z'}^\pi(s)$, then the gradient of $I_{\mathrm{SD3}}(s, z)$ to $\rho_{z^c}$ becomes

$$\nabla_{\rho_{z^c}} I_{\mathrm{SD3}}(s, z) = -1/(\lambda d_z^\pi(s) + \rho_{z^c}(s)), \quad (2)$$

where $I_{\mathrm{SD3}}(s, z)$ is the density ratio for a specific $(s, z)$ and the proof is attached in A.1. Thus, for skill $z$, increasing $\lambda$ will weaken the gradient of SD3 in reducing the state densities of other skills, which prevents skill collapse in SD3. (ii) We introduce explicit intra-skill exploration based on the latent space learned in estimating the skill density, which will be discussed in §3.2. To maximize $I_{\mathrm{SD3}}$, we adopt a modified Conditional Variational Auto-Encoder (CVAE) to stably estimate the state density for skills, which we introduce as follows.

**CVAE for State Density Estimation**  In SD3, we adopt a lower bound of skill-conditional state density (i.e., $\log d_z^\pi(s)$) via stochastic gradient variational Bayes. We adopt CVAE with a latent representation $h$ to obtain a variational form as

$$\log d_z^\pi(s) = \mathbb{E}_{Q(h|s,z)} \log \left[ P(s|z) \right] = \mathbb{E}_{Q(h|s,z)} \log \left[ \frac{P(s, h|z)}{Q(h|s, z)} \right] + \mathbb{E}_{Q(h|s,z)} \log \left[ \frac{Q(h|s, z)}{P(h|s, z)} \right]$$

$$\geq \mathbb{E}_{Q(h|s,z)} \log \left[ \frac{P(s|h, z)P(h|z)}{Q(h|s, z)} \right] = \underbrace{\mathbb{E}_{Q(h|s,z)} \log \left[ P(s|h, z) \right] - D_{\mathrm{KL}} \left[ Q(h|s, z) \| P(h|z) \right]}_{\mathcal{L}_z^{\mathrm{elbo}}(s)},$$

$$(3)$$

where the latent vector $h$ is sampled from a variational posterior distribution (i.e., $Q(h|s, z)$) conditioned on the state and skill, and the inequality holds by dropping off the non-negative second

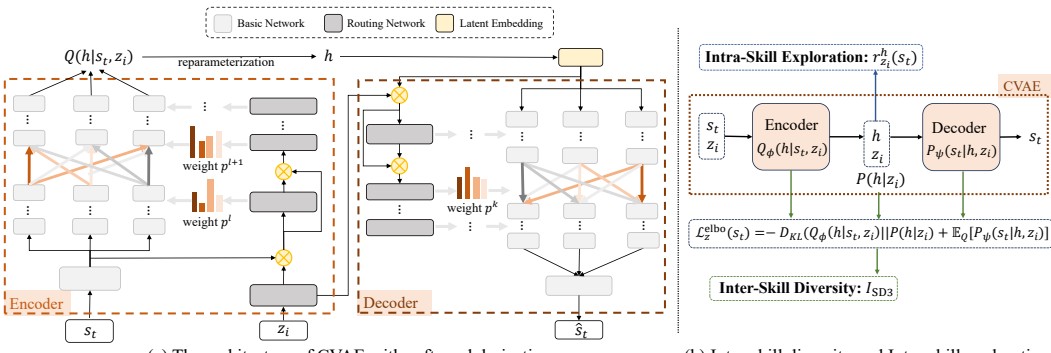

(a) The architecture of CVAE with soft modularization   (b) Inter-skill diversity and Intra-skill exploration

Figure 1: An overview of the CVAE architecture. (a) The encoder-decoder network with soft modularization. The feature extractor of state can be MLPs or convolution layers according to state- or image-based environment. (b) The inter-skill diversity objective for skill discovery and the intra-skill intrinsic reward for exploration can be derived from the learned CVAE.

term, which is the definition of $D_{\mathrm{KL}}(Q(h|s,z)\|P(h|s,z))$. Meanwhile, we use $P(s,h|z) = P(h|z)P(s|h,z)$ to decompose the joint distribution. According to Eq. (3), maximizing the Evidence Lower-Bound (ELBO) $\mathcal{L}_z^{\mathrm{elbo}}(s)$ can approximate the skill-conditioned state distribution, as $\log d_z^\pi(s) \approx \max_Q \mathcal{L}_z^{\mathrm{elbo}}(s)$. To maximize $\mathcal{L}_z^{\mathrm{elbo}}(s)$, we learn an encoder network $Q_\phi(h|s,z)$ to obtain the posterior of latent representation, where the posterior is represented by a diagonal Gaussian. Then, a latent vector $h$ is sampled from the posterior, and a decoder network $P_\psi(s|h,z)$ is used to reconstruct the state. The KL-divergence in $\mathcal{L}_z^{\mathrm{elbo}}(s)$ regularizes the latent space via a prior distribution $P(h|z)$, which is set to a standard Gaussian. The whole objective is optimized via stochastic gradient ascent with a reparameterization trick (Kingma & Welling, 2013; Kingma et al., 2019). To calculate $I_{\mathrm{SD3}}$, we perform state density estimations for all skills via forward inference based on the learned encoder and decoder. In calculating $I_{\mathrm{SD3}}$, we adopt efficient parallelization to calculate $\mathcal{L}_z^{\mathrm{elbo}}(s)$ for all skills $z \in \mathcal{Z}$ in one forward pass, which minimizes the run-time increase with the number of skills.

**Soft Modularization for CVAE**  As we maximize the state-density deviation in skill discovery, the resulting skills become diverse, and the corresponding state occupancy for different skills tends to be very different. In CVAE-based density estimation, since different skills share the same network parameters, optimizing $\mathcal{L}_z^{\mathrm{elbo}}$ for one skill can negatively affect the density estimation of other skills with significantly different state densities. Empirically, we also find obtaining an accurate estimation of $d_z^\pi(s)$ for all skills $z \in \mathcal{Z}$ can be difficult. As a result, we adopt a soft modularization technique that automatically generates soft network module combinations for different skills without explicitly specifying structures. As shown in Figure 1, the soft modularized CVAE contains an unconditional basic network and a routing network, where the routing network takes the skill and state embedding as input to estimate the routing strategy. Suppose each layer of the encoder/decoder network has $m$ modules, then the routing network gives the probabilities $p \in \mathbb{R}^{m \times m}$ to weight modules contributing to the next layer. Specifically, considering $l$-th layer has probabilities $p^l \in \mathbb{R}^{m \times m}$, then the probability in the next layer is

$$p^{l+1} = \mathcal{W}^l\big(\mathrm{ReLU}(g(p^l) \odot (u \odot v))\big), u = f_1(s), v = f_2(z), \qquad (4)$$

where $\odot$ denotes element-wise product, $g(\cdot)$, $f_1(\cdot)$ and $f_2(\cdot)$ are all fully connected layers that $f_1(\cdot)$ and $f_2(\cdot)$ map state $s$ and skill $z$ to the same dimensions (e.g., $d$), and $g(\cdot)$ maps $p^l$ to the dimension $d$. Then we have $\mathcal{W}^l \in \mathbb{R}^{m^2 \times d}$ to project the joint feature to a probability vector of layer $l+1$. In the basic network, we denote the input feature for the $j$-th module in the $l$-the layer as $g_j^l \in \mathbb{R}^d$; then we have $g_i^{l+1} = \sum_j \hat{p}_{i,j}^l(\mathrm{ReLU}(\mathcal{W}_j^l g_j^l))$ for the next layer, where $\hat{p}_{i,j}^l = \exp(p_{i,j}^l)/(\sum_{j=1}^m \exp(p_{i,j}^l))$ is the normalized vector that weights the $j$-th module in the $l$-th layer to contribute to the $i$-th module in the $l+1$-th layer. We remark that the soft modularization technique was originally proposed in multi-task RL (Yang et al., 2020), while we extend it to encoder-decoder-based CVAE for density estimation. The detailed architecture is given in §B.2.

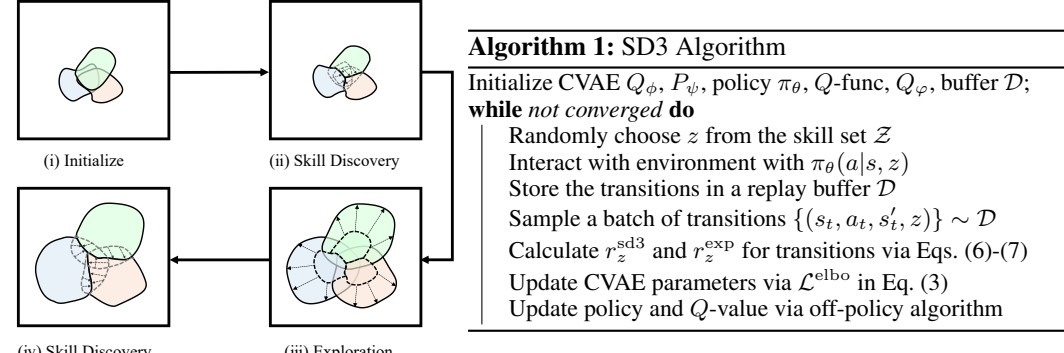

**Algorithm 1:** SD3 Algorithm

Initialize CVAE $Q_\phi$, $P_\psi$, policy $\pi_\theta$, $Q$-func, $Q_\varphi$, buffer $\mathcal{D}$;
**while** *not converged* **do**
  Randomly choose $z$ from the skill set $\mathcal{Z}$
  Interact with environment with $\pi_\theta(a|s,z)$
  Store the transitions in a replay buffer $\mathcal{D}$
  Sample a batch of transitions $\{(s_t, a_t, s'_t, z)\} \sim \mathcal{D}$
  Calculate $r_z^{\text{sd3}}$ and $r_z^{\text{exp}}$ for transitions via Eqs. (6)-(7)
  Update CVAE parameters via $\mathcal{L}^{\text{elbo}}$ in Eq. (3)
  Update policy and $Q$-value via off-policy algorithm

(i) Initialize    (ii) Skill Discovery

(iv) Skill Discovery    (iii) Exploration

Figure 2: An illustration of skill discovery in SD3. The skills start with overlapping areas and are separated via state-density deviation. Then, each skill explores the environment independently, resulting in overlapped but expanded areas. SD3 separates the areas again and leads to distinguishable skills. Such a process repeats and ultimately leads to exploratory and diverse skills.

## 3.2 LATENT SPACE EXPLORATION

As we discussed above, the SD3 objective that only maximizes the density deviation may lead to skill collapse. In addition to introducing an additional parameter $\lambda$ in Eq. (1), we find the learned CVAE in Figure 1 can provide a *free-lunch* intrinsic reward for efficient intra-skill exploration. In SD3, we derive an intrinsic reward based on the latent space that learns skill-conditioned representations for states. Specifically, the KL-divergence term $D_{\text{KL}}\big[Q(h|s,z)\|r(h)\big]$ in CVAE objective serves as an upper bound of the conditional MI term $I(S;H|Z)$, as

$$I(S;H|Z) = \mathbb{E}_{p(s,z),Q_\phi(h|s,z)}\big[\log Q_\phi(h|s,z)/P(h|z)\big] \leq \mathbb{E}_{p(s,z),Q_\phi(h|s,z)}\big[\log Q_\phi(h|s,z)/r(h)\big], \tag{5}$$

where $H$ denotes the random variable of the sampled latent representation $h$, and $r(h)$ the prior distribution set to a standard Gaussian, and $P(h|z) \triangleq \mathbb{E}_{P(s|z)}Q_\phi(h|s,z)$. The inequality holds since $D_{\text{KL}}[P(h|z)\|r(h)] \geq 0$ for all $z \in \mathcal{Z}$. Since $D_{\text{KL}}[Q_\phi(h|s,z)\|r(h)]$ is constrained in CVAE learning, the MI between states and latent representations for each skill is also compressed according to Eq. (5). Thus, the latent space in CVAE learns a compressive representation while retaining important information as the representation is then used for reconstruction. Based on the learned representation, we define the intrinsic reward for intra-skill exploration as

$$r_z^{\text{exp}}(s) = D_{\text{KL}}[Q_\phi(h|s,z)\|r(h)], \tag{6}$$

where $Q_\phi(h|s,z)$ is the posterior network learned in CVAE. The intrinsic reward in Eq. (6) quantifies the degree of compression of representation with respect to the state, which measures skill-conditioned state novelty in a compact space for intra-skill exploration. Intuitively, if a state $s^{(1)}$ is frequently visited by skill $z$, then the corresponding latent distribution is close to $r(h)$ according to Eq. (5), and the resulting reward $r_z^{\text{exp}}(s^{(1)})$ will be close to zero. In contrast, if a state $s^{(2)}$ is novel for skill $z$, then the corresponding intrinsic reward will be high since the latent posterior $Q_\phi(h|s^{(2)},z)$ can be very different from the prior $r(h)$. Thus, in exploration, such reward encourages the policy to find the scarcely visited states $\{s^+\}$ (with a high $D_{\text{KL}}[Q_\phi(h|s,z)\|r(h)]$) and explore these states.

An illustration of the skill learning process of SD3 is shown in Figure 2. The state occupancy of different skills overlaps initially in Figure 2(i), then we maximize $I_{\text{SD3}}$ via per-instance estimation and set it to an intrinsic reward as

$$r_z^{\text{sd3}}(s) = \log \frac{\lambda \, d_z^\pi(s)}{\lambda \, d_z^\pi(s)p(z) + \sum_{z' \neq z} d_{z'}^\pi(s)p(z')}, \tag{7}$$

which encourages skill density deviation and leads to more diverse skills with separate state coverage, as in Figure 2(ii). Then the exploration reward $r_z^{\text{exp}}(s)$ is used to encourage intra-skill exploration, which makes each skill explore unknown areas independently. After exploration, the state coverage of each skill increases and may lead to state-coverage overlapping again among skills, as in Figure 2(iii). Then the density-derivation reward $r_z^{\text{sd3}}(s)$ re-separates the updated areas to obtain distinguished skills, as in Figure 2(iv). The above process repeats for many rounds and SD3

finally learns exploratory and diverse skills. The algorithmic description of our method is given in Algorithm 1.

### 3.3 QUALITATIVE ANALYSIS

In this section, we give a qualitative analysis of the proposed SD3 objective and exploration reward, which encourage inter-skill diversity and intra-skill exploration, respectively.

The skill discovery objective $I_{\mathrm{SD3}}$ in Eq. (1) leads to diverse skills with separate explored areas, which is similar to the MI-based skill discovery objectives. As we usually set $\lambda \geq 1$ to prevent skill collapse, the following theorem connects $I_{\mathrm{SD3}}$ and the previous MI objectives.

**Theorem 3.1.** *With $\lambda \geq 1$, we have*

$$I(S; Z) \leq I_{\mathrm{SD3}} \leq c_0 + I(S; Z). \tag{8}$$

*where $c_0 = \log \lambda$. Specially, $I_{\mathrm{SD3}} = I(S; Z)$ if $\lambda = 1$.*

The above theorem shows when we maximize skill deviation via $I_{\mathrm{SD3}}$, the MI between $S$ and $Z$ also increases. The previous MI objective becomes a special case of $I_{\mathrm{SD3}}$, where the introduced $\lambda$ provides flexibility to control the strength of skill deviation. In the following, we connect the proposed intrinsic reward to the provably efficient count-based exploration in tabular cases.

Note that since $\lambda$ only relates to the overall objective $I_{\mathrm{SD3}}$ and does not affect the estimation of state density, the exploration bonus holds for arbitrary $\lambda \geq 1$.

**Theorem 3.2.** *In tabular MDPs, optimizing the intra-skill exploration reward is equivalent to count-based exploration, as*

$$r_z^{\exp}(s) \approx \frac{|\mathcal{S}|/2}{N(s, z) + \kappa}. \tag{9}$$

*where $N(s, z)$ is the count of visitation of state-skill pair $(s, z)$ in experiences, $|\mathcal{S}|$ is the total number of states in a tabular case, and $\kappa > 0$ is a small non-negative constant.*

As a result, maximizing the intra-skill exploration reward is equivalent to performing count-based exploration in previous works (Kolter & Ng, 2009; Strehl & Littman, 2008), which is provable efficient in tabular MDPs (Bellemare et al., 2016; Ostrovski et al., 2017). Through the approximation in a compact latent space, the intra-skill exploration encourages skill-conditional policy to increase the pseudo-count of rarely visited state-skill pairs in a high-dimensional space.

## 4 RELATED WORK

**Unsupervised Skill Discovery**   Unsupervised skill discovery in RL aims to acquire a repertoire of useful skills without relying on extrinsic rewards. Early efforts, such as VIC (Gregor et al., 2016), DIAYN (Eysenbach et al., 2019), and DADS (Sharma et al., 2020), maximize the MI between the skill and the state to discover diverse skills. However, as noted in EDL (Campos et al., 2020), LSD (Park et al., 2022), and CSD (Park et al., 2023), such MI-based methods usually prefer static skills caused by poor state coverage and may hinder the application for downstream tasks. Recent methods strive to address this limitation to learn dynamic and meaningful skills. These methods perform explicit exploration or enforce Lipschitz constraints in the representation to maximize the traveled distances of skills. Further, CIC (Laskin et al., 2022) employs contrastive learning between state transitions and skills to encourage agent's diverse behaviors. BeCL (Yang et al., 2023) uses contrastive learning to differentiate between various behavioral patterns and maximize the entropy implicitly. ReST (Jiang et al., 2022) encourages the trained skill to stay away from the estimated state visitation distributions of other skills. Some methods, like DISCO-DANCE (Kim et al., 2023), APS (Liu & Abbeel, 2021a), SMM (Lee et al., 2020) and DISDAIN (Strouse et al., 2022), focus on introducing an auxiliary exploration reward to address insufficient exploration. Furthermore, to verify the effectiveness of skill discovery in large-scale state space (e.g., images), recent methods including Choreographer (Mazzaglia et al., 2023) and Metra (Park et al., 2024) evaluate the effectiveness of methods on pixel-based URLB (Rajeswar et al., 2023), which often relies on model-based agents to learn meaningful knowledge from imagination, and skills are discovered in the latent space. Metra (Park et al., 2024) constructs a latent space associated with the original state space via a temporal

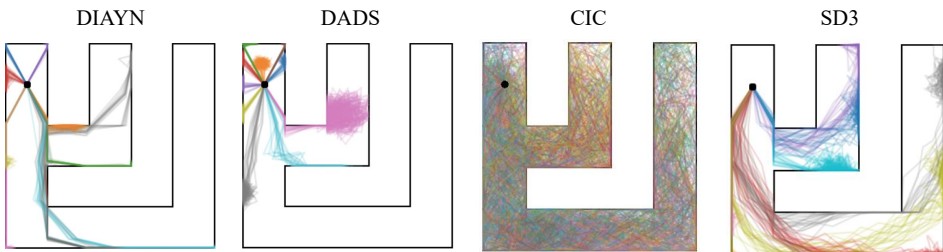

Figure 3: Results for maze experiment. We visually demonstrate the agent's ability to explore the environment and the diversity of skills discovered by the agent. The agent starts from the black dot of the maze and interacts for 250K steps. Both DIAYN and DADS do not reach the right side of the maze while obtaining distinguishable trajectories highlighted by different colors. The trajectories of CIC span the entire maze but appear chaotic. In contrast, SD3 can reach the farthest position from the starting point and facilitates easy differentiation of trajectories of different skills.

distance metric, which enables skill learning in high-dimensional environments by maximizing the coverage. In contrast, our method promotes skill diversity by encouraging deviations in skill density and enhances state coverage through latent space exploration. We validate our approach's efficacy through experiments on state-based and pixel-based tasks across various environments.

**Unsupervised RL** According to URLB (Laskin et al., 2021), URL algorithms are classified into three main categories: knowledge-based, data-based, and competence-based. Knowledge-based algorithms (Pathak et al., 2017; 2019; Burda et al., 2019) leverage the agent's predictive capacity or understanding of the environment, and the intrinsic reward is tied to the novelty of the agent's behaviors, encouraging the agent to explore areas where its model is less certain. Data-based algorithms (Liu & Abbeel, 2021b; Yarats et al., 2021) maximize the state entropy to maximize state coverage of skills. Competence-based algorithms (Lee et al., 2020; Eysenbach et al., 2019; Liu & Abbeel, 2021a; Nieto et al., 2021) pre-train the agent to learn useful skills that can be utilized to complete downstream tasks. Our method can be categorized as competence-based, while also combining the benefit of knowledge-based algorithms to encourage exploration. In addition, some recent algorithms do not easily fit into these categories. For example, LCSD (Ju et al., 2024) establishes connections between skills, states, and linguistic instructions to guide task completion based on external language directives. DuSkill (Kim et al., 2024) utilizes a guided diffusion model to generate versatile skills beyond dataset limitations, thereby enhancing the robustness of policy learning across diverse domains. EUCLID (Yuan et al., 2023) improves downstream policy learning performance by jointly pre-training dynamic models and unsupervised exploration strategies. VGCRL (Choi et al., 2021) applies variational empowerment to learn effective state representations, thereby improving exploration.

## 5 EXPERIMENTS

We start by introducing experiments in Maze to visualize the skills. Subsequently, we validate the effectiveness of SD3 by conducting experiments on challenging tasks from the DeepMind Control Suite (DMC) (Tassa et al., 2018), with both state-based (Laskin et al., 2021) and pixel-based (Rajeswar et al., 2023) observations. Finally, we conduct ablation studies to demonstrate the factors that influence the effectiveness of SD3.

### 5.1 MAZE EXPERIMENT

We conduct experiments in a 2D maze to visually demonstrate the learned skills, as shown in Figure 3. The agent's initial state is represented by a black dot, with different colored lines indicating the trajectories corresponding to the different skills it has learned. The agent's state is the current positional information, and the actions represent the velocity and direction of movement. Building on this, we compare SD3 with two classical MI-based methods, DIAYN (Eysenbach et al., 2019) and DADS (Sharma et al., 2020), whose objectives correspond to the reverse form $\mathcal{H}(Z) - \mathcal{H}(Z|S)$ and the forward form $\mathcal{H}(S) - \mathcal{H}(S|Z)$ of the MI term $I(S; Z)$, respectively. Additionally, we com-

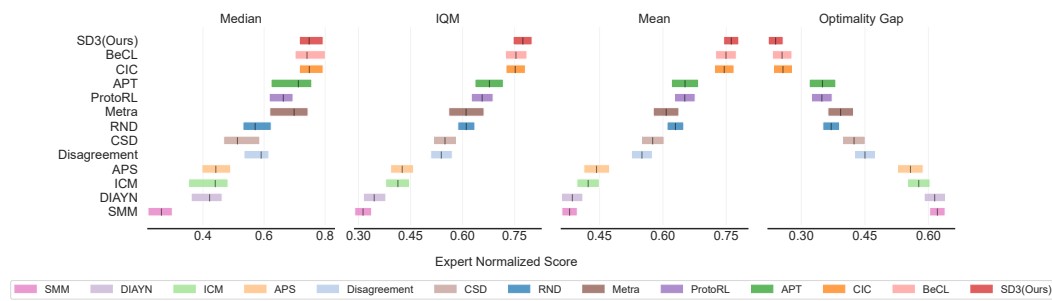

Figure 4: Results for state-based URLB. The aggregate statistics (Agarwal et al., 2021) indicate the adaptation performance of different unsupervised RL methods in 12 downstream tasks. In terms of IQM, Mean, and OG metrics, SD3 outperforms other competence-based methods and significantly surpasses pure exploration methods, achieving 77.37%, 76.19%, and 23.91%, respectively.

pare SD3 with an entropy-based CIC algorithm (Laskin et al., 2022), whose primary objective is to maximize state-transition entropy $\mathcal{H}(\tau)$ to generate diverse behaviors. We employ the PPO as the backbone and train $n = 10$ skills for each algorithm.

We delineate the learned skills of each algorithm within the maze environment in Figure 3 and introduce two key metrics for comparing SD3 with other methods: state coverage and distinguishability of skills, where insufficient state coverage may impede the acquisition of dynamic skills, and the lack of distinguishability leads to similar behaviors of skills. According to the results, (i) DIAYN and DADS fail to extend to the upper-right corner of the maze, but exhibit clear distinctions among trajectories of skills, indicating that merely maximizing $I(S; Z)$ can learn discriminable skills but lack effective exploration of the state space; (ii) CIC demonstrates the best state coverage while learns skills with mixed trajectories due to the maximization of $\mathcal{H}(s)$ as its primary objective; (iii) In contrast, SD3 strikes a balance between state coverage and empowerment in skill discovery. It learns discriminable skills by maximizing the deviation between the state densities of a certain skill and others. Meanwhile, SD3 achieves commendable state coverage through latent space exploration.

## 5.2 STATE-BASED URLB

According to state-based URLB (Laskin et al., 2021), we evaluate our approaches in 12 downstream tasks across 3 distinct continuous control domains, each designed to evaluate the effectiveness of algorithms under high-dimensional state spaces. The three domains are *Walker*, *Quadruped*, and *Jaco Arm*. Specifically, *Walker* involves a biped constrained to a 2D vertical plane with a state space $\mathcal{S} \in \mathbb{R}^{24}$ and an action space $\mathcal{A} \in \mathbb{R}^{6}$. The agent in the *Walker* domain must learn to maintain balance and move forward, completing four downstream tasks: *stand, walk, run,* and *flip*. *Quadruped* features a four-legged robot in a 3D environment, characterized by a state space $\mathcal{S} \in \mathbb{R}^{78}$ and an action space $\mathcal{A} \in \mathbb{R}^{16}$. The downstream tasks, including *stand, run, jump,* and *walk*, pose challenges to the agent due to the complex dynamics of its movements. *Jaco* employs a 6-DOF robotic arm with a three-finger gripper, functioning within a state space $\mathcal{S} \in \mathbb{R}^{55}$ and an action state $\mathcal{A} \in \mathbb{R}^{9}$. Primary downstream tasks in *Jaco* Arm include reaching and manipulating objects at various positions.

**Baselines.** We conduct comparisons between SD3 and the baselines delineated across the three URL algorithm categories as defined by URLB (Laskin et al., 2021). These categories encompass knowledge-based baselines, which consist of ICM (Pathak et al., 2017), Disagreement (Pathak et al., 2019), and RND (Burda et al., 2019); data-based baselines, which include APT (Liu & Abbeel, 2021b) and ProtoRL (Yarats et al., 2021); and competence-based baselines, comprising SMM (Lee et al., 2020), DIAYN (Eysenbach et al., 2019), and APS (Liu & Abbeel, 2021a). Furthermore, we extend our comparisons to include other novel competence-based algorithms such as CSD (Park et al., 2023), Metra (Park et al., 2024), BeCL (Yang et al., 2023), and CIC (Laskin et al., 2022).

**Evaluation.** We employ a rigorous evaluation to assess the performance of SD3 alongside other algorithms, involving a two-phase process. Initially, a pre-training of 2M steps is performed using only intrinsic rewards, followed by a fine-tuning phase of 100K steps on each downstream task using extrinsic rewards. Building upon prior work (Laskin et al., 2021), we utilize DDPG as the backbone

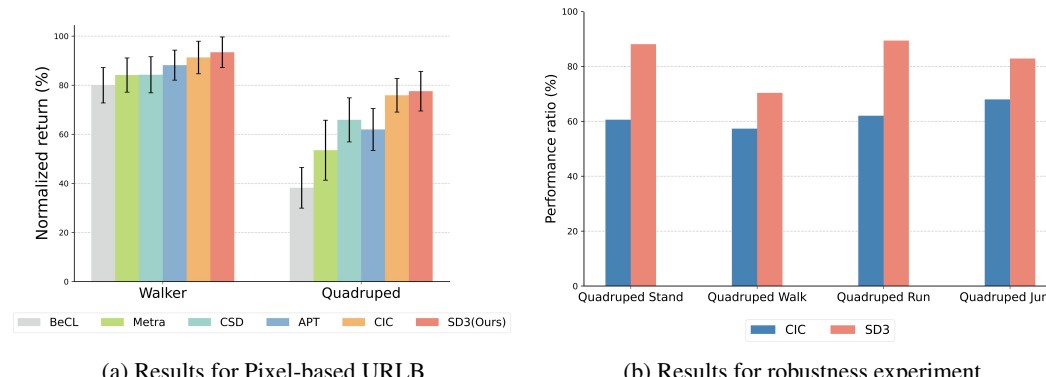

(a) Results for Pixel-based URLB  (b) Results for robustness experiment

Figure 5: (a) We conduct experiments on pixel-based URLB to demonstrate the scalability of SD3 for large-scale problems. (b) It can be observed that SD3 retains higher performance ratio than CIC in the noisy domain.

algorithm. To ensure statistical rigor and mitigate the impact of incidental factors in RL training, we conduct experiments across multiple seeds (10 seeds per algorithm), resulting in a substantial volume of experimental runs (i.e., $1560 = 13$ algorithms $\times$ 10 seeds $\times$ 3 domains $\times$ 4 tasks). We employ four statistical metrics to assess performance: Median, interquantile mean (IQM), Mean, and optimality gap (OG) (Agarwal et al., 2021). IQM focuses on the central tendency of the middle 50%, excluding the top and bottom quartiles. OG understands the extent to which the algorithm approaches the optimal level, where the optimal level is determined by the expert models' ultimate score obtained on each downstream task.

**Results.** According to Figure 4, SD3 achieves the highest IQM score at 77.37%, slightly surpassing CIC and BeCL, which scores 75.19% and 75.38% respectively, and significantly outperforming other competence-based algorithms such as Metra (61.01%), CSD (54.93%), and APS (43.61%). On the OG metric, SD3's gap to optimal performance is 23.91%, marginally better than CIC and BeCL at 25.65% and 25.44%, respectively, and far superior to Metra (39.25%), CSD (42.43%), and APS (55.76%). Additionally, compared to purely exploratory methods, SD3 significantly outperforms the best-performing method, APT, on both IQM and OG metrics, with APT scoring 67.74% and 34.98% on these metrics, respectively. The remarkable performance of SD3 stems from two main factors. First, the use of $r^{\mathrm{sd3}}$ facilitates the learning of distinguishable skills by the agent, thereby facilitating effective adaptation across various downstream tasks. Second, the learned compressed representation of the high-dimensional state space leads to efficient intra-skill exploration within a compact space, which not only maintains skill consistency but also enhances exploration ability.

### 5.3 PIXEL-BASED URLB

To further validate the effectiveness of SD3, we conduct experiments on pixel-based URLB (Rajeswar et al., 2023), which includes *Walker* and *Quadruped* domains with 8 downstream tasks. The pixel-based environment employs raw pixel data as input, foregoing abstracted features, or processed sensor information. The challenge of deriving meaningful skills from such unrefined inputs is substantial, particularly in the absence of external rewards. Meanwhile, exploration becomes more difficult in image-based spaces, thereby testing the exploration ability of algorithms under conditions that closely resemble practical applications.

**Baselines.** We compared SD3 with the top three performing algorithms in state-based experiments, i.e., BeCL (Yang et al., 2023), CIC (Laskin et al., 2022), and APT (Liu & Abbeel, 2021b), as well as with the recently proposed skill discovery algorithms including CSD (Park et al., 2023) and Metra (Park et al., 2024). Among these, APT stands out as a data-based algorithm, which can also be considered a representative of pure exploration algorithms and demonstrates strong performance in exploring environments. The others are competence-based algorithms, which accomplish downstream tasks by learning useful and diverse skills.

**Evaluation.** We conduct 2M steps of pre-training solely based on intrinsic rewards in each domain, followed by 100K steps of fine-tuning on the downstream tasks using extrinsic rewards. The

scores achieved in the downstream tasks are used to evaluate the algorithm. According to the official benchmark of the pixel-based URLB (Rajeswar et al., 2023), unsupervised RL algorithms often perform poorly when combined with a model-free method (e.g., DDPG (Lillicrap et al., 2016) or DrQv2 (Yarats et al., 2022)) with image observations, while performing much better when using a model-based backbone (e.g., Dreamer (Hafner et al., 2021)). Thus, we follow this setting and conduct experiments with Dreamer backbone. We report the average adaptation performance in Figure 5(a). In the relatively simple *Walker* domain, SD3 achieves the best performance (93.42%), slightly outperforming other methods (i.e., CIC-91.29%, APT-88.17%, CSD-84.26%). In the challenging Quadruped domain, SD3 outperforms CIC (77.57% and 75.89%, respectively) and shows significant improvement over other competence-based methods (i.e., CSD-65.89%, Metra-53.53%) and the best pure-exploration method in state-based URLB (i.e., APT-61.96%). This highlights SD3's commendable advantages in both various image-based tasks.

## 5.4 ROBUSTNESS EXPERIMENT

Unlike CIC, APS, and BeCL, which rely on entropy-based exploration strategies, SD3 introduces a novel exploration reward that resembles a UCB-style bonus. Such a UCB-term in exploration is provable efficient in linear and tabular MDPs, which has been rigorously studied in previous research (Jin et al., 2023; ZHANG et al., 2021). In contrast, the entropy-based exploration used in previous methods has the disadvantage of being non-robust (e.g., adding small noise will significantly affect its entropy). Thus, to further verify that the robustness of SD3, we conduct experiments in noisy domains of URLB by adding noise during pre-training, which is sampled from $N(0, 0.1)$, followed by noise-free fine-tuning to assess the learned skills.

**Evaluation.** We choose CIC for comparison, which performs competitively with our method in standard URLB. Each technique is evaluated across 5 random seeds and the results are given in Figure 5(b). The Performance Ratio (PR) denotes the ratio of the adaptation score in the noisy domain to that in the normal setup. According to the results, it is evident that the UCB-bonus used in SD3 is more robust than entropy-based rewards in noisy environments, achieving significantly higher Performance Ratio than CIC. The detailed results are attached in Appendix E.

## 5.5 ABLATION STUDIES AND VISUALIZATION

We provide ablation studies for components in skill discovery and skill adaptation of SD3. For skill discovery, we perform the comparison on (i) density estimation with and without soft modularization, and (ii) the different settings of temperatures in the routing network. The final rewards for skill discovery contain $r_z^{\mathrm{sd3}}(s)$ and $r_z^{\mathrm{exp}}(s)$. We conduct ablation studies on (iii) different settings of $\lambda$ in calculating $r_z^{\mathrm{sd3}}(s)$, as well as (iv) the different balance factors of the two rewards. For skill adaptation, we sampled skills randomly to evaluate their generalization ability in our main results. In ablation studies, (v) we evaluate two more skill-choosing strategies in adaptation for a comparison. We refer to Appendix D for detailed results and analysis. We also provide visualizations of skills learned in tree-like Maze and DMC tasks in Appendix C. The results show that SD3 learns dynamic and valuable skills, enabling the agent to adapt to downstream tasks quickly.

## 6 CONCLUSION

We propose a novel skill discovery method that promotes skill diversity by encouraging skill deviations in state density and enhancing state coverage through latent space exploration. We realize a novel soft modularization architecture for state density estimation of different skills. Theoretically, the skill discovery objective also maximizes the initial MI term, and the resulting intra-skill exploration bonus resembles count-based exploration. Moreover, our four experiments complement each other and collectively provide sufficient evidence that SD3 demonstrates superior and more comprehensive performance compared to other methods. One limitation of our method is that the soft modularization architecture is limited to discrete skill spaces, and the theoretical analysis of the exploration bonus requires the assumption of tabular MDPs. In the future, we will extend the idea of skill discovery to LLM-based agents to learn meaningful skills in more complex environments.

## 7 ETHICS STATEMENT

This work does not involve any human subjects or personally identifiable information. This work focuses on the field of unsupervised RL, and therefore does not involve any datasets. No sensitive data or unethical methodologies are employed. We declare no conflicts of interest related to the sponsorship or publication of this work. The research has adhered to the ICLR Code of Ethics, with special attention to fairness and bias concerns.

## 8 REPRODUCIBILITY

All experiments and results reported in this paper can be reproduced using the provided anonymous source code. Details regarding the model architecture and training parameters are included in Appendix B. The theorems discussed in section 3.3 are supported by detailed proofs provided in Appendix A.

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

# A    THEORETICAL PROOFS

## A.1    PROOF OF EQ. 2

*Proof.* As we discussed in 3.1, since we uniformly sample skills from the skill set that contains $n$ skills, we have $p(z) = 1/n$ for each skill. Then we have

$$I_{\text{SD3}}(s, z) = \log \frac{\lambda n d_z^\pi(s)}{\lambda d_z^\pi(s) + \sum_{z' \neq z} d_{z'}^\pi(s)} = \log \frac{\lambda n d_z^\pi(s)}{\lambda d_z^\pi(s) + \rho_{z^c}(s)}. \tag{10}$$

Then, the gradient of $I_{\text{SD3}}(s, z)$ to $\rho_{z^c}(s)$ becomes

$$\nabla_{\rho_{z^c}} I_{\text{SD3}}(s, z) = \frac{\lambda d_z^\pi(s) + \rho_{z^c}(s)}{\lambda n d_z^\pi(s)} \frac{-\lambda n d_z^\pi(s)}{\left(\lambda d_z^\pi(s) + \rho_{z^c}(s)\right)^2}$$

$$= -\frac{1}{\lambda d_z^\pi(s) + \rho_{z^c}(s)}. \tag{11}$$

This completes the proof. □

## A.2    PROOF OF THEOREM 3.1

*Proof.* For clarity, we write $I_{\text{SD3}}$ as $I_{\text{SD3}}(\lambda)$ to explicitly highlight its dependency on the parameter $\lambda$ in the following context. We first note that the function $I_{\text{SD3}}(\lambda)$ is monotonically increasing relative to $\lambda$, and $I_{\text{SD3}}(\lambda)$ is equal to $I(S; Z)$ when $\lambda = 1$. Therefore, the first inequality

$$I(S; Z) \leq I_{\text{SD3}}(\lambda)$$

always holds for $\lambda \geq 1$. Next, it remains to prove the second inequality, which suffices to give an upper bound of $I_{\text{SD3}}(\lambda) - I(S; Z)$. Note that

$$I_{\text{SD3}}(\lambda) - I(S; Z)$$

$$= \mathbb{E}_{z \sim p(z), s \sim d_z^\pi(s)} \left[ \log \frac{\lambda \, d_z^\pi(s)}{\lambda \, d_z^\pi(s) p(z) + \sum_{z' \neq z} d_{z'}^\pi(s) p(z')} \cdot \frac{d_z^\pi(s) p(z) + \sum_{z' \neq z} d_{z'}^\pi(s) p(z')}{d_z^\pi(s)} \right]$$

$$= \log \lambda + \mathbb{E}_{z \sim p(z), s \sim d_z^\pi(s)} \left[ \log \frac{d_z^\pi(s) p(z) + \sum_{z' \neq z} d_{z'}^\pi(s) p(z')}{\lambda \, d_z^\pi(s) p(z) + \sum_{z' \neq z} d_{z'}^\pi(s) p(z')} \right]$$

$$= \log \lambda - \mathbb{E}_{z \sim p(z), s \sim d_z^\pi(s)} \left[ \log \frac{d_z^\pi(s) p(z) + \sum_{z' \neq z} d_{z'}^\pi(s) p(z') + (\lambda - 1) d_z^\pi(s) p(z)}{d_z^\pi(s) p(z) + \sum_{z' \neq z} d_{z'}^\pi(s) p(z')} \right]$$

$$= \log \lambda - \mathbb{E}_{z \sim p(z), s \sim d_z^\pi(s)} \left[ \log \left( 1 + (\lambda - 1) \frac{d_z^\pi(s) p(z)}{d_z^\pi(s) p(z) + \sum_{z' \neq z} d_{z'}^\pi(s) p(z')} \right) \right]. \tag{12}$$

Recalling that $d_z^\pi(\cdot)$ denotes the state density of skill $z$, and $p(z)$ is the probability density function of skill $z$, we know that the term

$$\log \left( 1 + (\lambda - 1) \frac{d_z^\pi(s) p(z)}{d_z^\pi(s) p(z) + \sum_{z' \neq z} d_{z'}^\pi(s) p(z')} \right) \tag{13}$$

is always non-negative for $\lambda \geq 1$. Therefore, we have

$$I_{SD3}(\lambda) \leq \log \lambda + I(S; Z). \tag{14}$$

This completes the proof. $\qquad\square$

### A.3 PROOF OF THEOREM 3.2

*Proof.* In this proof, we first give a formulation of the intrinsic reward in a linear parameterized assumption, and then discuss the special case of tabular MDPs.

With linear assumptions, we denote $\eta(s_t, z_t) \in \mathbb{R}^d$ as the feature vector of $(s_t, z_t)$, which is extracted by the *encoder* network of CVAE. The decoder network is assumed to be a linear function of the feature vector as $\hat{s}_t = W_t \eta(s_t, z_t)$, where $W_t \in \mathbb{R}^{c \times d}$ and $\hat{s}_t \in \mathbb{R}^c$. Then the reconstruction of the state becomes a regularized least-squared problem that captures the prediction error given a dataset $\mathcal{D}_m$, where $m$ is the number of episodes in the dataset. Thus, we have

$$W_t = \arg\min_W \sum_{i=0}^{m} \left\| s_t^i - W\eta(s_t^i, z_t^i) \right\|_F^2 + \kappa \cdot \|W\|_F^2, \tag{15}$$

where $\|\cdot\|_F$ denotes the Frobenius norm. We further define the following noise with respect to the least-square problem in Eq. (15) as

$$s_t = W_t \eta(s_t, z_t) + \epsilon, \quad \epsilon \sim \mathcal{N}(0, \mathbf{I}). \tag{16}$$

Here we consider the estimation error $\epsilon$ in Eq. (15) to follow the standard multivariate Gaussian distribution.

Recall that our practical intra-skill exploration reward is $D_{\mathrm{KL}}[Q_\phi(h|s, z) \| r(h)]$, where $Q_\phi$ is a posterior network compressing the representation of each state and skill with parameter $\phi$, and $r(h)$ is the marginal distribution of the latent variable, where we follow previous works (Alemi et al., 2017; Bai et al., 2021) to consider the marginal as the standard normal distribution. Then we redefine the intrinsic reward in a Bayesian perspective, where we introduce $\Phi$ to denote the total parameters, as

$$r_z^{\mathrm{exp}}(s) = \mathbb{E}_\Phi D_{\mathrm{KL}}[Q_\phi(h|s, z) \| r(h)] = \mathcal{H}(Q^{\mathrm{margin}}) - \mathcal{H}(Q_\phi(h|s, z)), \tag{17}$$

where $Q^{\mathrm{margin}} = Q(s, z)|\mathcal{D}_m$ is the margin distribution of the encoding over the posterior of the parameters $\Phi$. In practice, we replace the expectation over posterior $\Phi$ by the corresponding point estimation, namely the parameter $\phi$ of the neural networks trained with SD3 model on the dataset $\mathcal{D}_m$. Formally, considering the Bayesian form of learning objective, we have

$$r_z^{\mathrm{exp}}(s) = \mathcal{H}(Q^{\mathrm{margin}}) - \mathcal{H}(Q_\phi(h|s, z)) = \mathcal{H}(Q(s, z, S)|\mathcal{D}_m) - \mathcal{H}(Q(s, z, S)|\Phi, \mathcal{D}_m), \tag{18}$$

where $Q$ is a neural network in practice. We adopt the mapping

$$Q(s, z, S)|\Phi, \mathcal{D}_m = Q_\phi(h|s, z) \tag{19}$$

since $Q_\phi$ is trained to reconstruct the variable $S$, where $\phi$ constitutes a part of the parameters of the total parameters $\Phi$. According to Data Processing Inequality, the post-processing of the signal does not increase information, and we can understand $Q$ as post-processing mapping the state-skill vector via an encoder network. Then we have the following inequality for the information-gain term:

$$\begin{aligned} r_z^{\mathrm{exp}}(s) &= \mathcal{H}(Q(s, z, S)|\mathcal{D}_m) - \mathcal{H}(Q(s, z, S)|\Phi, \mathcal{D}_m) \\ &\leq \mathcal{H}(s, z, S|\mathcal{D}_m) - \mathcal{H}(s, z, S|\Phi, \mathcal{D}_m) = I(\Phi; (s, z, S)|\mathcal{D}_m), \end{aligned} \tag{20}$$

where we denote $(s, z)$ as realizations as they are sampled from the dataset as input, and $S$ is a random variable that is learned to reconstruct by parameter $\Phi$. The inequality can be tight since $Q(\cdot)$ is trained by reconstruction, which contains sufficient information about $(s, z, S)$.

In the following, we will prove the following inequality in a linear case with a parameter $W_t$ considered in Eq. (15), as

$$r_{z_t}^{\mathrm{exp}}(s_t) \leq I(W_t; (s_t, z_t, S_t)|\mathcal{D}_m) \leq \frac{c}{2}[\eta(s_t, z_t)^\top \Lambda_t^{-1} \eta(s_t, z_t)], \tag{21}$$

the $[\eta(s_t, z_t)^\top \Lambda_t^{-1} \eta(s_t, z_t)]$ term is known as an upper-confidence-bound (UCB)-term in linear MDPs (Jin et al., 2023; Cai et al., 2020), and $\Lambda_t = \sum_{j=1}^{m} \eta(s_j, z_j)\eta(s_j, z_j)^\top + \kappa \cdot \mathbf{I}$ is the covariance matrix of the samples in the dataset. Finally, we will connect the UCB-term to the count-based bonus in the tabular case.

Let denote $\mathrm{vec}(W_t)$ as vectorization of $W_t \in \mathbb{R}^{c \times d}$, and also $\tilde{\eta}(s_t, z_t)$

$$
\mathrm{vec}(W_t) = \begin{bmatrix} w_{11} \\ \vdots \\ w_{1d} \\ w_{21} \\ \vdots \\ w_{2d} \\ \vdots \\ \vdots \\ w_{c1} \\ \vdots \\ w_{cd} \end{bmatrix} \in \mathbb{R}^{cd}, \quad \tilde{\eta}(s_t, z_t) = \begin{bmatrix} \eta(s_t, z_t) & 0 & \cdots & 0 \\ 0 & \eta(s_t, z_t) & \cdots & 0 \\ \vdots & \vdots & \ddots & \vdots \\ 0 & 0 & \cdots & \eta(s_t, z_t) \end{bmatrix} = \begin{bmatrix} \eta_1 & 0 & \cdots & 0 \\ \vdots & & & \\ \eta_d & 0 & \cdots & 0 \\ & \eta_1 & \cdots & 0 \\ \vdots & \vdots & & \vdots \\ & \eta_d & \cdots & 0 \\ \vdots & \vdots & & \vdots \\ \vdots & \vdots & & \vdots \\ 0 & 0 & \cdots & \eta_1 \\ \vdots & \vdots & & \vdots \\ 0 & 0 & \cdots & \eta_d \end{bmatrix} \in \mathbb{R}^{cd \times c},
$$

$$(22)$$

then it is not difficult to verify that $\mathrm{vec}(W_t)^\top \tilde{\eta}(s_t, z_t) = W_t \eta(s_t, z_t)$. By the definition of the mutual information, we observe

$$
\begin{aligned}
I(W_t; [s_t, z_t, S_t] \mid \mathcal{D}_m) &= I(\mathrm{vec}(W_t); [s_t, z_t, S_t] \mid \mathcal{D}_m) \\
&= \mathcal{H}(\mathrm{vec}(W_t) \mid \mathcal{D}_m) - \mathcal{H}(\mathrm{vec}(W_t) \mid \mathcal{D}_m \cup (s_t, z_t, S_t)) \\
&= \frac{1}{2} \log \det \left( \mathrm{Var}(\mathrm{vec}(W_t) \mid \mathcal{D}_m) \right) - \frac{1}{2} \log \det \left( \mathrm{Var}(\mathrm{vec}(W_t) \mid \mathcal{D}_m \cup (s_t, z_t, S_t)) \right).
\end{aligned}
$$

$$(23)$$

Next, we need to obtain $\mathrm{Var}(\mathrm{vec}(W_t) \mid \mathcal{D}_m)$ and $\mathrm{Var}(\mathrm{vec}(W_t) \mid \mathcal{D}_m \cup (s_t, z_t, S_t))$. Recalling that $\epsilon$ satisfies the standard Gaussian distribution in Eq. (16), we can conclude that

$$
s_t | \eta_t, W_t \sim \mathcal{N}(\mathrm{vec}(W_t)^\top \tilde{\eta}(s_t, z_t), \mathbf{I}).
$$

Assuming the prior distribution $W \sim \mathcal{N}(0, \mathbf{I}/\kappa)$, then the prior of $\mathrm{vec}(W)$ also follows from $\mathcal{N}(0, \mathbf{I}/\kappa)$. Moreover, using Bayes' theorem and plugging the probability of $p(\mathrm{vec}(W_t))$, we have

$$
\begin{aligned}
\log p(\mathrm{vec}(W_t) \mid \mathcal{D}_m) &= \log p(\mathrm{vec}(W_t)) + \log p(\mathcal{D}_m \mid \mathrm{vec}(W_t)) - \log p(\mathcal{D}_m) \\
&= -\|\mathrm{vec}(W_t)\|^2/2 - \sum_{i=1}^{m} \|\mathrm{vec}(W_t)\tilde{\eta}(s_t^i, z_t^i) - s_{t+1}^i\|^2/2 + \mathrm{Const} \\
&= -(\mathrm{vec}(W_t) - \tilde{\mu}_{t,m})^\top \tilde{\Lambda}_{t,m}^{-1}(\mathrm{vec}(W_t) - \tilde{\mu}_{t,m})/2 + \mathrm{Const},
\end{aligned}
$$

$$(24)$$

where $\tilde{\mu}_t$ and $\tilde{\Lambda}_t$ in the last equality are defined as

$$
\tilde{\mu}_{t,m} = \tilde{\Lambda}_t^{-1} \sum_{i=0}^{m} \tilde{\eta}(s_t^i, z_t^i) s_{t+1}^i \in \mathbb{R}^{cd}, \qquad \tilde{\Lambda}_{t,m} = \sum_{i=0}^{m} \tilde{\eta}(s_t^i, z_t^i)\tilde{\eta}(x_t^i, z_t^i)^\top + \kappa \cdot \mathbf{I} \in \mathbb{R}^{cd \times cd}.
$$

Taking the left-hand side of $\log$ to the right. The Eq. (24) implies the distribution of $\mathrm{vec}(W_t) \mid \mathcal{D}_m \sim N(\tilde{\mu}_{t,m}, \tilde{\Lambda}_{t,m}^{-1})$. Hence, we can get

$$
\mathrm{Var}(\mathrm{vec}(W_t) \mid \mathcal{D}_m) = \tilde{\Lambda}_{t,m}^{-1}, \qquad \mathrm{Var}(\mathrm{vec}(W_t) \mid \mathcal{D}_m \cup (s_t, z_t, S_t)) = \tilde{\Lambda}_{t,m+1}^{-1}.
$$

$$(25)$$

We proceed to derive Eq. (23) by applying Eq. (25), from which we obtain

$$
\begin{aligned}
I(\mathrm{vec}(W_t); [s_t, z_t, S_{t+1}] | \mathcal{D}_m) &= \frac{1}{2} \log \det \left( \tilde{\Lambda}_{t,m}^{-1} \right) - \frac{1}{2} \log \det \left( \tilde{\Lambda}_{t,m+1}^{-1} \right) \\
&= \frac{1}{2} \log \det \left( \tilde{\Lambda}_{t,m+1} + \tilde{\eta}(s_t, z_t)\tilde{\eta}(s_t, z_t)^\top \right) - \frac{1}{2} \log \det \left( \tilde{\Lambda}_{t,m} \right) \\
&= \frac{1}{2} \log \det \left( \tilde{\eta}(s_t, z_t)^\top \tilde{\Lambda}_t^{-1} \tilde{\eta}(s_t, z_t) + \mathbf{I} \right),
\end{aligned}
$$

$$(26)$$

where the last equality holds by applying the Matrix Determinant Lemma to the first term. Recalling our definition of $\tilde{\eta}(s_t, z_t)$, the state-skill pairs are finite in the tabular case, so we have

$$
\begin{aligned}
\tilde{\Lambda}_t &= \sum_{i=0}^{m} \tilde{\eta}(s_t^i, z_t^i)\tilde{\eta}(x_t^i, z_t^i)^\top + \kappa \cdot \mathbf{I} \\
&= \begin{bmatrix}
\sum \eta(s_0,z_0)\eta(s_0,z_0)^\top + \kappa I & 0 & \cdots & 0 \\
0 & \sum \eta(s_1,z_1)\eta(s_1,z_1)^\top + \kappa I & \cdots & 0 \\
\vdots & \vdots & \ddots & \vdots \\
0 & 0 & \cdots & \sum \eta(s_m,z_m)\eta(s_m,z_m)^\top + \kappa I
\end{bmatrix}.
\end{aligned} \tag{27}
$$

Then, $\tilde{\eta}(s_t, z_t)^\top \tilde{\Lambda}_t^{-1} \tilde{\eta}(s_t, z_t)$ can be rewritten as

$$
\begin{aligned}
&\tilde{\eta}(s_t, z_t)^\top \tilde{\Lambda}_t^{-1} \tilde{\eta}(s_t, z_t) \\
&= \begin{bmatrix}
\eta(s_t,z_t)^\top & 0 & \cdots & 0 \\
0 & \eta(s_t,z_t)^\top & \cdots & 0 \\
\vdots & \vdots & \ddots & \vdots \\
0 & 0 & \cdots & \eta(s_t,z_t)^\top
\end{bmatrix}
\begin{bmatrix}
\Lambda^{-1} & 0 & \cdots & 0 \\
0 & \Lambda^{-1} & \cdots & 0 \\
\vdots & \vdots & \ddots & \vdots \\
0 & 0 & \cdots & \Lambda^{-1}
\end{bmatrix}
\begin{bmatrix}
\eta(s_t,z_t) & 0 & \cdots & 0 \\
0 & \eta(s_t,z_t) & \cdots & 0 \\
\vdots & \vdots & \ddots & \vdots \\
0 & 0 & \cdots & \eta(s_t,z_t)
\end{bmatrix} \\
&= \begin{bmatrix}
\eta(s_t,z_t)^\top \Lambda^{-1} \eta(s_t,z_t) & 0 & \cdots & 0 \\
0 & \eta(s_t,z_t)^\top \Lambda^{-1} \eta(s_t,z_t) & \cdots & 0 \\
\vdots & \vdots & \ddots & \vdots \\
0 & 0 & \cdots & \eta(s_t,z_t)^\top \Lambda^{-1} \eta(s_t,z_t)
\end{bmatrix} \in \mathbb{R}^{c \times c}.
\end{aligned} \tag{28}
$$

Therefore, by eliminating the determinant based on the expression in Eq. (28) and applying the inequality $\log(1 + x) \le x$ for $x \ge 0$, we can further bound Eq (26) from above as

$$
\begin{aligned}
I(\text{vec}(W_t); [s_t, z_t, S_{t+1}]|\mathcal{D}_m) &= \frac{1}{2} \cdot \log \det \left( \tilde{\eta}(s_t, z_t)^\top \tilde{\Lambda}_t^{-1} \tilde{\eta}(s_t, z_t) + \mathbf{I} \right) \\
&= \frac{c}{2} \cdot \log \left( \eta(s_t, z_t)^\top \Lambda^{-1} \eta(s_t, z_t) + 1 \right) \\
&\le \frac{c}{2} \cdot \eta(s_t, z_t)^\top \Lambda^{-1} \eta(s_t, z_t).
\end{aligned} \tag{29}
$$

Hence, based on Eq. (20) and Eq. (29), we conclude that

$$
r_z^{\exp}(s_t) \le I(W_t; [s_t, z_t, S_{t+1}]|\mathcal{D}_m) = I(\text{vec}(W_t); [s_t, z_t, S_{t+1}]|\mathcal{D}_m) \le \frac{c}{2} \cdot \eta(s_t, z_t)^\top \Lambda^{-1} \eta(s_t, z_t). \tag{30}
$$

In tabular cases (Auer & Ortner, 2006), the state and skill are considered as finite and countable. Let $d = |\mathcal{S}| \times |\mathcal{Z}|$. Recall that $\eta(s_t, z_t) \in \mathcal{R}^{|\mathcal{S}||\mathcal{Z}|}$ is the one-hot vector with a value of 1 at position $(s_t, z_t) \in \mathcal{S} \times \mathcal{Z}$, i.e.,

$$
\eta(s_j, z_j) = \begin{bmatrix} 0 \\ \vdots \\ 1 \\ \vdots \\ 0 \end{bmatrix} \in \mathbb{R}^d, \quad \text{and} \quad \eta(s_j, z_j)\eta(s_j, z_j)^\top = \begin{bmatrix} 0 & \cdots & 0 & \cdots & 0 \\ \vdots & \ddots & & & \vdots \\ 0 & & 1 & & 0 \\ \vdots & & & \ddots & \vdots \\ 0 & \cdots & 0 & \cdots & 0 \end{bmatrix} \in \mathbb{R}^{d \times d}. \tag{31}
$$

We denote the gram matrix $\Lambda_j = \sum_{i=0}^{m} \eta(s_j^i, z_j^i)\eta(s_j^i, z_j^i)^\top + \kappa \cdot \mathbf{I}$ for $\kappa > 0$ as covariance matrix given a dataset $\mathcal{D}_m$. Since we denote $\eta$ as a one-hot vector, and $\Lambda$ as the sum of all the matrices $\eta(s_j, z_j)\eta(s_j, z_j)^\top$, each diagonal element of $\Lambda$ can be seen as the corresponding count $N(s_j, z_j)$ for the state-skill pair, i.e.

$$
\Lambda = \begin{bmatrix}
N(s_0,z_0)+\kappa & 0 & \cdots & 0 \\
0 & N(s_1,z_1)+\kappa & \cdots & 0 \\
\vdots & & \ddots & \vdots \\
0 & & N(s_j,z_j)+\kappa & 0 \\
\vdots & & & \ddots & \vdots \\
0 & \cdots & \cdots & N(s_m,z_m)+\kappa
\end{bmatrix}.
$$

Moreover, given a dataset, the expression on the right side of the theorem's inequality is inversely proportional to the total number of state-skill pairs; in other words,

$$\eta(s_j, z_j)^\top \Lambda_t^{-1} \eta(s_j, z_j) = \frac{1}{N(s_j, z_j) + \kappa}. \tag{32}$$

According to Eq. (21), we have the following relationship in the tabular case:

$$r_{z_t}^{\exp}(s_t) \leq I(W_t; (s_t, z_t, S_t)|\mathcal{D}_m) \leq \frac{c}{2}[\eta(s_t, z_t)^\top \Lambda_t^{-1} \eta(s_t, z_t)] = \frac{|\mathcal{S}|/2}{N(s_t, z_t) + \kappa}. \tag{33}$$

The first inequality is due to the Data Processing Inequality according to Eq. (20). The bound is tight since $Q(\cdot)$ is trained by reconstruction, which contains sufficient information about $(s, z, S)$. The second inequality is tight when $\eta(s_t, z_t)^\top \Lambda_t^{-1} \eta(s_t, z_t) \to 0$, which means that the count of state-action pair is large. In the last equation, $c$ is the count of all states in the tabular space. Thus, we have

$$r_z^{\exp}(s) \approx \frac{|\mathcal{S}|/2}{N(s, z) + \kappa}, \tag{34}$$

if the count of $N(s, z)$ is large. Intuitively, optimizing the reward $\eta(s, z)^\top \Lambda^{-1} \eta(s, z)$ incentivizes the agent to increase the visitation of $(s, z)$. Furthermore, since we have proven that Eq. (21) holds, we can state that in the tabular case, maximizing the intra-skill reward is equivalent to maximizing the count-based rewards (Bellemare et al., 2016; Ostrovski et al., 2017). The intra-skill exploration reward encourages the skill-conditional policy to increase the visitation times of those rare state-skill pairs.

$\square$

# B  HYPER-PARAMETERS AND IMPLEMENTATION DETAILS

## B.1  HYPER-PARAMETERS

We utilize the baselines from the open-source implementations of URLB (https://github.com/rll-research/url_benchmark), CIC (https://github.com/rll-research/cic), and BeCL (https://github.com/Rooshy-yang/BeCL), keeping their hyper-parameters fixed throughout both the pre-training and fine-tuning stages. For CSD and Metra, due to the absence of their experiments in state- and pixel-based URLBs, we re-implement them in these benchmarks on their official implementations (CSD https://github.com/seohongpark/CSD-locomotion, Metra https://github.com/seohongpark/METRA ). Table 1 details the hyper-parameters used for SD3 and DDPG.

## B.2  IMPLEMENTATION DETAILS

**Soft Modularized CVAE**  To achieve SD3, we utilize a soft modularized CVAE to estimate the state density $d_z^\pi(s)$ of one skill. Specifically, we forward the state $s$ through an MLP or CNN to obtain a $d$-dimensional state embedding $f_1(s)$ and, similarly, obtain a $d$-dimensional skill embedding $f_2(z)$. We use $f_1(s)$ as the input to the unconditional basic network, and $f_1(s) \odot f_2(z)$ as the input to the routing network. The basic network comprises $n$ layers, each containing $m$ modules, for progressively extracting features. The routing network contains $n - 1$ gating layers, which provide a probability vector $p^l$ based on the input as shown in Eq. (4) to weight the contribution of the $l$-th layer's modules to the $l + 1$-th layer's modules. Particularly, the probability vector which outputs from the first layer of the routing network is represented as

$$p^{l=1} = \mathcal{W}^l\big(\text{ReLU}(f_1(s) \odot f_2(z))\big). \tag{35}$$

Then, the probability vector is normalized using the softmax function as $\hat{p}^l$ and the input to each module in the basic network can be expressed as

$$g_i^{l+1} = \sum_j \hat{p}_{i,j}^l(\text{ReLU}(\mathcal{W}_j^l g_j^l)), \tag{36}$$

Table 1: Hyper-parameters used for SD3 and DDPG.

| SD3 hyper-parameter | Value |
| --- | --- |
| Skill dim | 16 discrete |
| Softmax Temperature $T$ | 1 |
| Skill sampling frequency (steps) | 50 |
| Exploration ratio $\alpha$ | $\{0.04, 2.0\}$ |
| Weight Parameter $\lambda$ | 1.5 |
| CVAE Encoder arch. | $\dim(S) \to 1024 \to 1024 \to 1024 \to 40 * 2$ ReLU (MLP) |
| CVAE Decoder arch. | $40 \to 1024 \to 1024 \to 1024 \to \dim(S)$ ReLU (MLP) |
| **DDPG hyper-parameter** | **Value** |
| Replay buffer capacity | $10^6$ |
| Action repeat | 1 |
| Seed frames | 4000 |
| $n$-step returns | 3 |
| Mini-batch size | 1024 |
| Seed frames | 4000 |
| Discount ($\gamma$) | 0.99 |
| Optimizer | Adam |
| Learning rate | $10^{-4}$ |
| Agent update frequency | 2 |
| Critic target EMA rate ($\tau_Q$) | 0.01 |
| Features dim. | 1024 |
| Hidden dim. | 1024 |
| Exploration stddev clip | 0.3 |
| Exploration stddev value | 0.2 |
| Number pretraining frames | $2 \times 10^6$ |
| Number finetuning frames | $1 \times 10^5$ |

where $\hat{p}_{i,j}^l$ weights the $j$-th module in the $l$-th layer to contribute to the $i$-th module in the $l+1$-th layer, $g_j^l$ is the input to the $j$-th module in the $l$-th layer and $\mathcal{W}_j^l$ represents the module parameters.

By progressively extracting features of state $s$ while incorporating the weight information, the encoder transforms the state $s$ into the mean $\mu(s|z)$ and variance $\sigma^2(s|z)$ of the latent space conditioned on the skill $z$. The latent representation $h$ is generated using the reparameterization trick, ensuring gradients can be backpropagated through the sampling process. Specifically, this is done as $h = \mu + \sigma \cdot \epsilon$, where $\epsilon$ is the noise sampled from a standard Gaussian distribution. The decoder then progressively up-samples and reconstructs the output state $\hat{s}$ from the latent representation $h$, incorporating the weight information generated by the routing network. We train the entire soft modularized CVAE by maximizing $\mathcal{L}^{\text{elbo}}$ as given in Eq. (3), enabling it to more accurately estimate $d_z^\pi(s)$.

**Practical Implementation** We propose the complete SD3 algorithm in Algorithm 2. We conduct our experiments using an RTX 4090 GPU. Each run in the state-based URLB environment takes approximately 1 day, while runs in the Maze environment requires about 3 hours each. For the pixel-based URLB environment, each run takes around 4 days or less.

## C VISUALIZATION

### C.1 TREE-LIKE MAZE

As shown in Figure 6, we conduct additional experiments in the tree-like maze to visualize the skills learned by SD3. It can be observed that DIAYN and DADS only reach the middle of the maze, whereas SD3 successfully reaches the bottom of the maze. The proposed latent space reward in SD3 demonstrates strong exploration ability in large-scale mazes. Moreover, the trajectories of different skills remain distinguishable in SD3.

---

**Algorithm 2:** Complete SD3 algorithm

---

**Input:** number of pre-training frames $N_{PT}$, number of fine-tuning frames $N_{FT}$, batch size $N$, skill sampling frequency $N_{\text{update}}$, skill set $\mathcal{Z}$, exploration ratio $\alpha$.
**Initialize** Environment, CVAE $Q_\phi$, actor $\pi_\theta$, critic $Q_\varphi$, replay buffer $\mathcal{D}$.

*//Pre-training*
**for** $t = 1$ **to** $N_{PT}$ **do**
    Randomly choose $z$ from the skill set $\mathcal{Z}$ every $N_{\text{update}}$ steps
    Interact with environment by $\pi_\theta(a|s,z)$
    Store the transition in replay buffer $\mathcal{D} \leftarrow \mathcal{D} \cup (s_t, a_t, s'_t, z)$.
    **if** $t \geq 4,000$ **then**
        Sample a batch from $\mathcal{D} : \{s_t, a_t, s'_t, z\}^N \sim \mathcal{D}$.
        Update CVAE $Q_\phi$ via $\mathcal{L}^{\text{elbo}}$ in Eq. (3).
        Use CVAE $Q_\phi$ to compute $d_z^\pi(s)$ and $d_{z' \neq z}^\pi(s)$.
        Compute $r_z^{\text{sd3}}(s)$ and $r_z^{\text{exp}}(s)$ with Eqs. (6)-(7).
        Compute the intrinsic reward $r^{\text{int}} = r_z^{\text{sd3}}(s) + \alpha \cdot r_z^{\text{exp}}(s)$
        Update actor $\pi_\theta$ and critic $Q_\varphi$ using intrinsic reward $r^{\text{int}}$.
    **end if**
**end for**

*//Fine-tuning*
**for** $t = 1$ **to** $N_{FT}$ **do**
    Use pre-training models to initialize actor $\pi_{\theta'}$ and critic $Q_{\varphi'}$.
    Randomly sample a skill $z^*$ from $\mathcal{Z}$ and fix the $z^*$.
    Interact with environment by $\pi_{\theta'}$.
    Store the transition in replay buffer $\mathcal{D} \leftarrow \mathcal{D} \cup (s_t, a_t, r_t, s'_t, z^*)$.
    **if** $t \geq 4,000$ **then**
        Sample a batch from $\mathcal{D} : \{s_t, a_t, r_t, s'_t, z^*\}^N \sim \mathcal{D}$.
        Use extrinsic reward $r_t$ obtained from downstream task to update $\pi_{\theta'}$ and $Q_{\varphi'}$.
    **end if**
**end for**

---

## C.2 DEEPMIND CONTROL SUITE

Figure 7 shows the learned skills in the *Walker*, *Quadruped*, and *Jaco Arm* domains. The result shows SD3 can learn various locomotion skills, including standing, walking, rolling, moving, and somersault; and also learns various manipulation skills by moving the arm to explore different areas, opening and closing the gripper in different locations. The learned meaningful skills lead to superior generalization performance in the fine-tuning stage of various downstream tasks.

# D ABLATION STUDIES

## D.1 THE EXPLORATION RATIO

We conduct an ablation on the different exploration ratios $\alpha$, Specifically, with the hyper-parameter $\alpha$, the reward is represented as:

$$r_z^{\text{total}}(s) = r_z^{\text{sd3}}(s) + \alpha \cdot r_z^{\text{exp}}(s). \tag{37}$$

As illustrated in Figure 8(a), when $\alpha$ is set to 0 and 0.02, the agent can learn distinguishable and convergent skills but fails to fully explore the maze. When $\alpha$ is set to 0.08, the agent explores sufficiently, but the trajectories at the endpoints are quite scattered, indicating that the learned skill strategies lack stability. In contrast, $\alpha = 0.04$ balances exploration and the skill diversity.

According to our analysis, when the proportion of exploration is deficient or even absent, SD3 solely maximize $I_{\text{SD3}}$. Conversely, an excessively high $\alpha$ can overly prioritize intra-skill exploration, resulting in instability within the learned skills. Empirically, we have found that $\alpha = 0.04$ can lead to promising results in downstream tasks in the Quadruped domain.

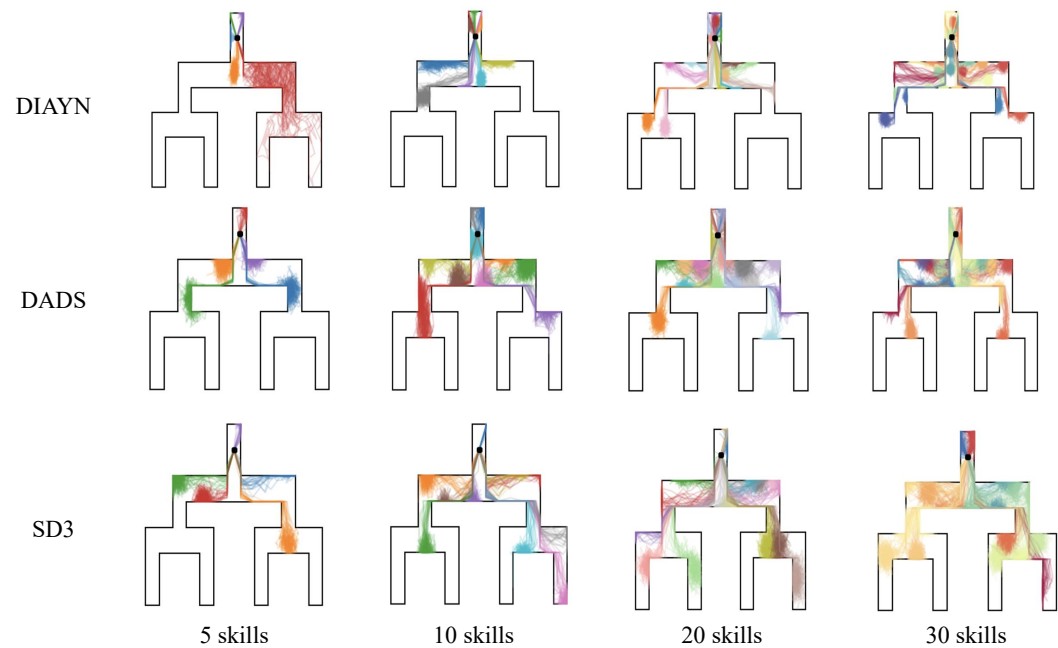

Figure 6: Additional experiments in the tree-like Maze with different numbers of skills. Under different environmental conditions, SD3 demonstrates superior exploration capabilities while still learning distinguishable skills, outperforming DIAYN and DADS.

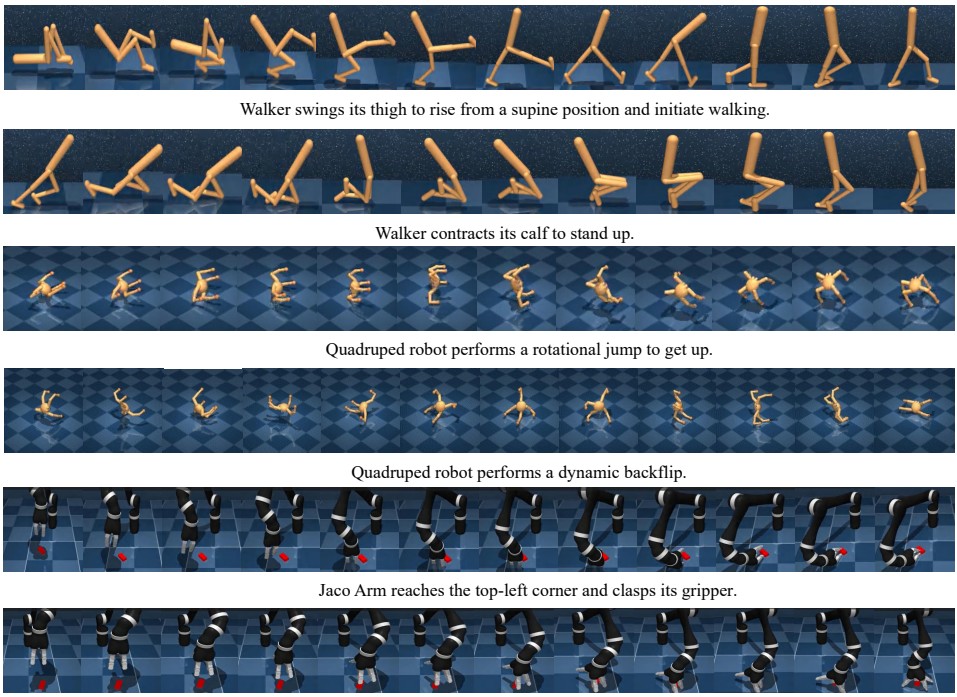

Figure 7: Skill visualization in DMC. It can be observed that SD3 learns dynamic and valuable skills, which enable the agent to quickly adapt to downstream tasks.

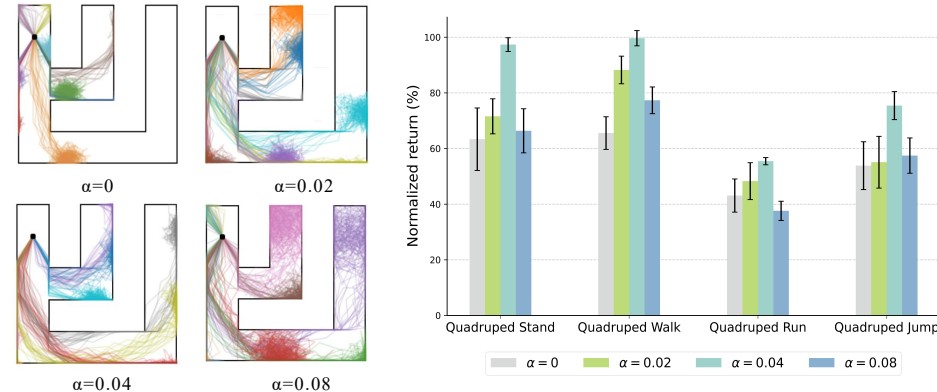

(a) The impact of exploration ratio in maze environment     (b) The impact of exploration ratio in state-based Quadruped

Figure 8: Results for the impact of exploration ratio. (a) We conduct experiments with different $\alpha$ in the maze and found that varying $\alpha$ values significantly impact both the state coverage and the stability of learned skills. (b) In the *Quadruped* domain, different $\alpha$ also have a notable effect on the performance of various downstream tasks.

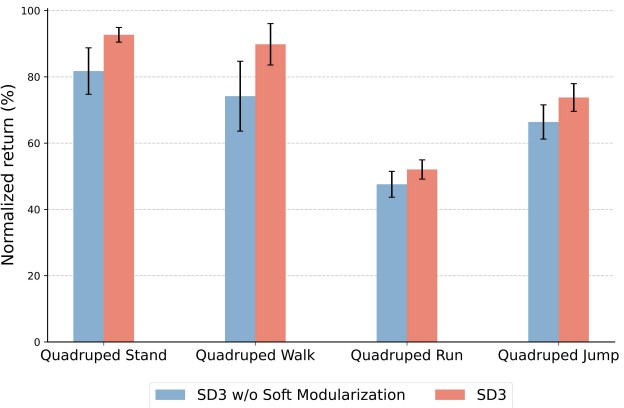

Figure 9: Ablation on the soft modularization structure.

### D.2 IMPACT OF SOFT MODULARIZATION

As mentioned in section 3.1, we use CVAE to estimate the state density of different skills. To enhance the accuracy of estimation in complex state spaces, we have introduced soft modularization into the traditional CVAE structure. Consequently, we conduct an ablation study on the soft modularization. Aggregated scores are reported in Figure 9. We observe that SD3 with soft modularized CVAE obtains superior performance, as it has sufficient capacity to learn the density information of different skills for the same state in complex state spaces, while the skill density estimation of one skill may intervene with those of other skills in the traditional CVAE.

### D.3 TEMPERATURE IN SOFTMAX

In section 3.1, we mentioned that the normalized weight $\hat{p}^l_{i,j}$ for the routing network is computed with the equation $\hat{p}^l_{i,j} = \exp(p^l_{i,j})/(\sum_{j=1}^m \exp(p^l_{i,j}))$. This is implemented using *Softmax*, where we follow the previous work (Hinton et al., 2015) to introduce a *temperature $T$* to control the level of uncertainty in output probabilities. The formula is as follows:

$$\hat{p}^l_{i,j} = \frac{\exp(p^l_{i,j}/T)}{\sum_{j=1}^m \exp(p^l_{i,j}/T)}. \tag{38}$$

From the above formula, it can be observed that when the temperature $T = 1$, it resembles the original softmax function. As $T$ decreases, the distribution output by softmax gradually becomes

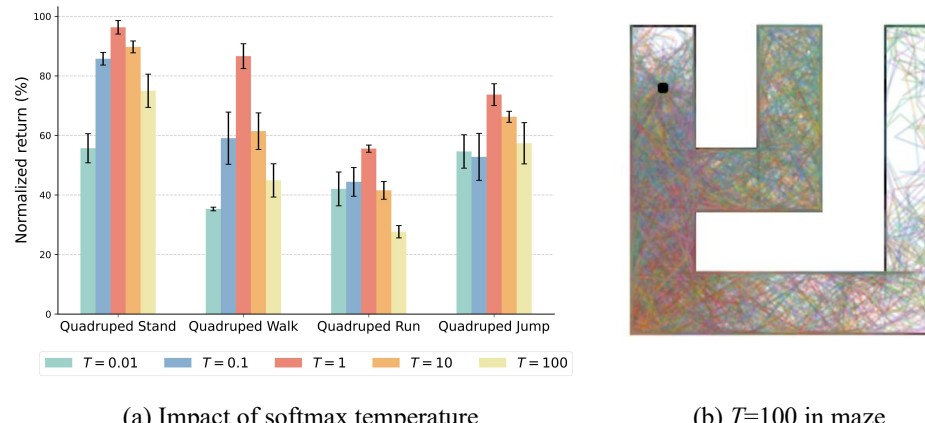

(a) Impact of softmax temperature  (b) $T$=100 in maze

Figure 10: Results for the impact of softmax temperature. (a) We exhibit the performance of the agent with different temperatures in the *Quadruped* domain. (b) When the temperature is set to 100, SD3 becomes a count-based pure exploration method. It demonstrates a certain degree of environment exploration capability but lacks empowerment in the environment.

more extreme, eventually converging to a deterministic distribution. Conversely, as $T$ increases, the softmax gradually tends to derive a uniform distribution. Here we perform the ablation study on the temperature coefficient.

The result is illustrated in Figure 10(a). When $T$ values are 0.1 and 0.01, the output of the softmax function in the routing network will gradually approximate $\operatorname{argmax}(p_i^l)$, at which point each training iteration utilizes only a single module from each layer. This practice inevitably diminishes the accuracy of estimating $d_z^\pi(s)$; when $T$ values are 10 and 100, the routing network tends to output uniformly distributed weight values, causing the routing network to fail. The entire network structure can be approximated as a VAE composed of multiple modules. Given the loss of skill $z$ information in the basic network, the intrinsic reward of SD3 can be repsented as follows:

$$
\begin{aligned}
r_z^{\text{total}}(s) &= r_z^{\text{sd3}}(s) + r_z^{\text{exp}}(s) \\
&= \log \frac{\lambda\, d^\pi(s)}{\lambda\, d^\pi(s)p(z) + \sum_{z' \neq z} d^\pi(s)p(z')} + D_{\text{KL}}[Q_\phi(\cdot|s)\|r(h)] \\
&= \log \frac{\lambda \cdot n}{\lambda + n - 1} + D_{\text{KL}}[Q_\phi(\cdot|s)\|r(h)] \\
&= c + D_{\text{KL}}[Q_\phi(\cdot|s)\|r(h)],
\end{aligned}
\tag{39}
$$

where $c$ represents a constant. Furthermore, based on Theorem 3.2, the right-hand side of the equation can be approximated as $\frac{|\mathcal{S}|/2}{N(s)+\kappa}$. Substituting into Eq. (39), we can obtain:

$$
r_z^{\text{total}}(s) \approx c + \frac{|\mathcal{S}|/2}{N(s) + \kappa}.
\tag{40}
$$

At this point, the SD3 method transforms into a count-based exploration approach. As presented in Figure 10(b), the agent learns extremely dynamic behavior, thereby preventing it from adequately adapting downstream tasks.

### D.4 IMPACT OF WEIGHT PARAMETER $\lambda$

The discussion in section 3.1 introduces a weight parameter $\lambda$ in Eq.(1). To investigate the impact of $\lambda$, we conduct an ablation study by varying $\lambda$ from $[0.5, 1.0, 1.5, 2.0, 3.0]$. The results, exhibited in Figure 11, indicate that the performance of SD3 fluctuates within a narrow range when lambda is greater than 1. Therefore, we conclude that $\lambda$ is generally applicable in a wider range, and SD3 is not sensitive to the parameter when $\lambda >= 1.5$.

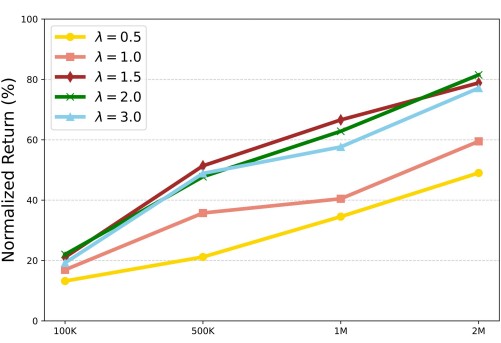 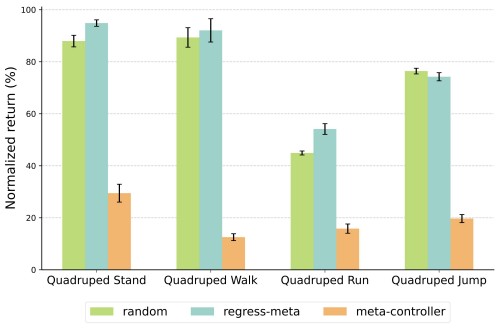

Figure 11: Results for the impact of weight parameter in the *Quadruped*. When $\lambda$ is set to 0.5 or 1, SD3 performs poorly. However, it is observed that increasing lambda beyond 1 does not significantly impact the performance of SD3.

Figure 12: Skill adaption strategies ablation. We test several adaptation methods in the fine-tuning phase and find that randomly selecting skills perform comparably to using regress-meta, but employing the meta-controller results in a decline in the performance.

### D.5 SKILL ADAPTION STRATEGIES IN FINE-TUNING

Previous work (Laskin et al., 2021) has shown that during the fine-tuning phase, performance across different skills does not always level equally; some skills demonstrate weaker adaptability in downstream tasks, while others show the opposite. Therefore, we investigate various skill adaptation methods in the state-based environment to assess their impact on algorithm performance in downstream tasks.

In the experiment described in section 5.2, for a fair comparison, we adhere to the standards set in the URLB, employing a random sampling skills method during the fine-tuning stage to evaluate the average performance of skills. Therefore, here we introduce two additional skill adaptation methods: *regress-meta* and *meta-controller*. Regress-meta computes the expected reward for each skill during the first 4K steps of the fine-tuning phase to determine its skill-value, and then selects the skill with the highest skill-value to perform the downstream task. Meta-controller trains an upper-level controller $\mu(z|s)$ in the fine-tuning phase to select the most appropriate skill for the current state $s$, thereby combining it with the policy $\pi(a|s, z)$ trained in the pre-training phase and optimizing the high-level policy based on $\pi(a|s) = \sum_{z \in \mathcal{Z}} \mu(z|s)\pi(a|s, z)$.

Results are shown in Figure 12. The performance of using regress-meta to select skills shows improvements compared to randomly selecting skills in *Quadruped Stand*, *Walk*, and *Run* but a slight drop in *Quadruped Jump*. We attribute this to the fact that regress-meta consistently selects the skill with the highest expected reward during the initial steps of the fine-tuning phase. While this approach does increase the probability of choosing a skill with good adaptability, there is also a risk of choosing a skill that performs well during the initial 4K steps but exhibits mediocre performance thereafter. In contrast, the meta-controller exhibits relatively poor performance. We hypothesize that the meta-controller usually requires a large number of examples to train, which is difficult to converge within the 100K fine-tuning steps.

## E NUMERICAL RESULT

In Table 2 and Table 3, we present the mean normalized scores and standard errors of all algorithms across 12 downstream tasks within the state-based URLB experiments. SD3 demonstrates superior performance across multiple downstream tasks. In Table 4, we present the results of the pixel-based URLB experiments. Across 8 downstream tasks, SD3 displays notable competitiveness compared to other baselines. Additionally, we showcase the results of robustness experiments in Table 5.

Table 2: Results of SD3 and novel competence-based methods on state-based URLB.

| Domain | Task | DDPG | CSD | Metra | CIC | BeCL | SD3 |
|---|---|---|---|---|---|---|---|
| Walker | Flip | 538±27 | 615±17 | 600±48 | **641±26** | **611±18** | 595±25 |
| | Run | 325±25 | 445±13 | 302±23 | **450±19** | 387±22 | **451±23** |
| | Stand | 899±23 | **962±7** | 951±7 | **959±2** | 952±2 | 930±5 |
| | Walk | 748±47 | 857±51 | 756±67 | **903±21** | 883±34 | **914±11** |
| Quadruped | Jump | 236±48 | 357±39 | 300±9 | 565±44 | **727±15** | 676±29 |
| | Run | 157±31 | 362±60 | 276±20 | 445±36 | **535±13** | 471±13 |
| | Stand | 392±73 | 455±36 | 637±36 | 700±55 | **875±33** | 847±17 |
| | Walk | 229±57 | 224±18 | 200±27 | 621±69 | **743±68** | 752±40 |
| Jaco | Reach bottom left | 72±22 | 99±7 | 143±9 | **154±6** | 148±13 | **151±7** |
| | Reach bottom right | 117±18 | 106±6 | 142±8 | **149±4** | 139±14 | **152±9** |
| | Reach top left | 116±22 | 101±7 | 130±13 | **149±10** | 125±10 | **142±7** |
| | Reach top right | 94±18 | 154±11 | 158±16 | 163±9 | 126±10 | 152±7 |

Table 3: Results of other baselines on state-based URLB.

| Domain | Task | ICM | Disagreement | RND | APT | ProtoRL | SMM | DIAYN | APS |
|---|---|---|---|---|---|---|---|---|---|
| Walker | Flip | 390±10 | 332±7 | 506±29 | 606±30 | 549±21 | 500±28 | 361±10 | 448±36 |
| | Run | 267±23 | 243±14 | 403±16 | 384±31 | 370±22 | 395±18 | 184±23 | 176±18 |
| | Stand | 836±34 | 760±24 | 901±19 | 921±15 | 896±20 | 886±18 | 789±48 | 702±67 |
| | Walk | 696±46 | 606±51 | 783±35 | 784±52 | 836±25 | 792±42 | 450±37 | 547±38 |
| Quadruped | Jump | 205±47 | 510±28 | 626±23 | 416±54 | 573±40 | 167±30 | 498±45 | 389±72 |
| | Run | 125±32 | 357±24 | 439±7 | 303±30 | 324±26 | 142±28 | 347±47 | 201±40 |
| | Stand | 260±45 | 579±64 | 839±25 | 582±67 | 625±76 | 266±48 | 718±81 | 435±68 |
| | Walk | 153±42 | 386±51 | 517±41 | 582±67 | 494±64 | 154±36 | 506±66 | 385±76 |
| Jaco | Reach bottom left | 88±14 | 117±9 | 102±9 | 143±12 | 118±7 | 45±7 | 20±5 | 84±5 |
| | Reach bottom right | 99±8 | 122±5 | 110±7 | 138±15 | 138±8 | 60±4 | 17±5 | 94±8 |
| | Reach top left | 80±13 | 121±14 | 88±13 | 137±20 | 134±7 | 39±5 | 12±5 | 74±10 |
| | Reach top right | 106±14 | 128±11 | 99±5 | **170±7** | 140±9 | 32±4 | 21±3 | 83±11 |

# F  MORE DISCUSSIONS

## F.1  THE UNIQUE FAVORABLE PROPERTIES OF SD3

Previous skill discovery methods, such as CIC, APS, and BeCL, also encourage exploration while discovering diverse skills. However, in comparison, SD3 possesses its own distinctive properties.

First, SD3 introduces a new objective for skill discovery, which is not derived from maximizing MI. The core principle of SD3 is to promote deviation in exploration regions across different skills, thereby facilitating more effective skill discovery. Unlike previous methods focusing on maximizing a lower-bound of MI, SD3 uses a novel CVAE architecture for density estimation to directly estimate the original objective. Further, as shown in Theorem 3.1, a qualitative analysis reveals that the previous MI objective is merely a special case of SD3.

Second, different from APS, CIC, and BeCL, which explicitly or implicitly maximizes state entropy for exploration, SD3 adopts a novel exploration strategy that resembles count-based exploration. In section 5.4, we confirm that such UCB-style reward is more robust than entropy-based reward. Meanwhile, this exploration reward can be estimated as an byproduct in from the learned CVAE, avoiding the additional mechanisms compared to other methods.

## F.2  THE PERFORMANCE OF SD3 COMPARED TO CIC

The quantitative results in Figs 4 and 5(a) indicate that SD3 and CIC are comparable. While SD3 slightly outperforms CIC, the improvement may not be statistically significant. In fact, in the experimental section of the main text, our focus is on showcasing SD3's overall performance and advantages.

Regarding the skill discovery objective, we believe that evaluating the fine-tuning performance of skills is somewhat limited. As demonstrated in the maze experiment (see Figure 3), although CIC achieves the best state coverage, it learns very disorganized skills with mixed trajectories. While CIC attains high scores after fine-tuning, it fails to reflect the core objective of skill discovery, which aims to learn diverse and distinguishable skills. SD3, on the other hand, excels in discovering easily distinguishable skills and also demonstrates competitive performance in downstream tasks.

Table 4: Results of SD3 and baselines on pixel-based URLB.

| Domain | Task | APT | CSD | Metra | CIC | BeCL | SD3 |
|--------|------|-----|-----|-------|-----|------|-----|
| Walker | Flip | 803±26 | 681±56 | 665±32 | **836±12** | 539±8 | **864±27** |
|  | Run | 506±4 | 451±41 | 454±29 | 504±21 | 456±14 | **543±22** |
|  | Stand | 961±5 | 958±13 | 968±4 | **973±2** | 968±4 | **982±1** |
|  | Walk | 880±37 | 948±5 | 949±3 | **953±5** | 939±1 | **945±3** |
| Quadruped | Jump | 557±67 | 580±74 | 677±27 | **723±16** | 340±32 | **729±16** |
|  | Run | 396±9 | 390±21 | 276±46 | **439±3** | 162±4 | **438±16** |
|  | Stand | 785±18 | 854±20 | 788±29 | 873±13 | 583±56 | **921±3** |
|  | Walk | 475±55 | 530±19 | 181±39 | **672±15** | 283±39 | **680±43** |

Table 5: Results of robustness experiments.

| Task | CIC (Noisy) | CIC (Normal) | Performance Ratio | SD3 (Noisy) | SD3 (Normal) | Performance Ratio |
|------|-------------|--------------|-------------------|-------------|--------------|-------------------|
| walker_flip | 511±6 | 641±26 | 79.72% | 554±24 | 595±25 | **93.11%** |
| walker_run | 319±20 | 450±19 | **70.89%** | 330±25 | 451±23 | **73.17%** |
| walker_stand | 845±12 | 959±2 | 88.11% | 909±11 | 930±5 | **97.74%** |
| walker_walk | 784±46 | 903±21 | 86.82% | 877±27 | 914±11 | **95.95%** |
| quad_jump | 384±61 | 565±44 | 67.96% | 560±48 | 676±29 | **82.84%** |
| quad_run | 276±48 | 445±36 | 62.02% | 421±47 | 471±13 | **89.38%** |
| quad_stand | 424±25 | 700±55 | 60.57% | 746±93 | 847±17 | **88.07%** |
| quad_walk | 356±99 | 621±69 | 57.32% | 529±55 | 752±40 | **70.34%** |
| Average | – | – | 71.68% | – | – | **86.33%** |

Additionally, we conduct experiments to confirm that SD3 is more robust than CIC in noisy environments. Our four experiments in the main text complement each other and collectively provide sufficient evidence that SD3 demonstrates superior and more comprehensive performance compared to other methods, including the ability to discover distinguishable skills (i.e., in maze/URLB domains), superior performance in downstream tasks (i.e., in state/pixel URLB), and scalability to large-scale problems (i.e., pixel-based domains). Therefore, we believe that SD3 will be favored over CIC and other methods for a wide range of tasks.

### F.3 THE INTEGRATION OF EXPLORATION AND DIVERSITY REWARDS DURING TRAINING

As vividly displayed in Figure 2, to better explain the key idea behind our algorithm and to illustrate the skill discovery process of SD3, we describe the learning process in an iterative manner. However, in practice, we first obtain a combined intrinsic reward $r^{\text{int}} = r^{\text{sd3}} + \alpha r^{\text{exp}}$ of two objectives, and then adopt DDPG as a backbone RL algorithm to learn the policy. We adopt such an optimization approach because using a combined reward $r^{\text{int}}$ only requires learning a single $Q$-function, which is more computationally efficient than an iterative process that requires learning two $Q$-functions.

Table 6: Results of SD3 without soft-modularized CVAE.

| Task | CIC | BeCL | SD3 | SD3(w/o soft-modu) |
|------|-----|------|-----|---------------------|
| Quad Stand | 700 ± 55 | 875 ± 33 | 847 ± 17 | 752 ± 64 |
| Quad Walk | 621 ± 69 | 743 ± 68 | 752 ± 40 | 642 ± 80 |
| Quad Run | 445 ± 36 | 535 ± 13 | 471 ± 13 | 422 ± 34 |
| Quad Jump | 565 ± 44 | 727 ± 15 | 676 ± 29 | 589 ± 45 |
| Walker Stand | 959 ± 2 | 952 ± 2 | 930 ± 5 | 910 ± 16 |
| Walker Walk | 903 ± 21 | 883 ± 34 | 914 ± 11 | 870 ± 30 |
| Walker Run | 450 ± 19 | 387 ± 22 | 451 ± 23 | 409 ± 55 |
| Walker Flip | 641 ± 26 | 611 ± 18 | 595 ± 25 | 523 ± 33 |
| Jaco Top Left | 149 ± 10 | 125 ± 10 | 142 ± 7 | 125 ± 5 |
| Jaco Top Right | 163 ± 9 | 126 ± 10 | 152 ± 7 | 117 ± 5 |
| Jaco Bottom Left | 154 ± 6 | 148 ± 13 | 151 ± 7 | 134 ± 8 |
| Jaco Bottom Right | 149 ± 4 | 139 ± 14 | 152 ± 9 | 122 ± 8 |

Table 7: Results for different number of modules.

| Task | Quad Stand | Quad Walk | Quad Run | Quad Jump |
|------|------------|-----------|----------|-----------|
| Module = 2 | $777 \pm 26$ | $638 \pm 45$ | $377 \pm 38$ | $499 \pm 32$ |
| Module = 3 | $\mathbf{862 \pm 25}$ | $650 \pm 29$ | $456 \pm 26$ | $590 \pm 20$ |
| Module = 5 | $781 \pm 31$ | $\mathbf{799 \pm 31}$ | $390 \pm 32$ | $541 \pm 24$ |
| Module = 6 | $680 \pm 27$ | $323 \pm 32$ | $261 \pm 31$ | $375 \pm 34$ |
| Module = 4 | $847 \pm 17$ | $752 \pm 40$ | $\mathbf{471 \pm 13}$ | $\mathbf{676 \pm 29}$ |

# G ADDITIONAL ABLATION STUDY

## G.1 STATE-BASED URLB WITHOUT SOFT-MODULARIZED CVAE

We conduct experiments without the soft-modularized CVAE in state-based URLB, using a standard CVAE where the encoder consists of a 4-layer MLP network. The results, shown in the Table 6, demonstrate that even without the soft-modularized CVAE, our method still achieves competitive performance on several downstream tasks.

## G.2 THE NUMBER OF MODULES IN SOFT-MODULARIZED CVAE

We conduct ablation experiments on the number of modules in the state-based quadruped environment, and the results are shown in Table 7. From the table, it can be observed that the performance differences are minimal when the number of modules is set to 2, 3, or 5. However, when the number is increased to 6, there is a significant performance drop. We attribute this to the difficulty in effectively training the soft-modularized structure as the number of modules becomes too large. The relatively comprehensive performance is achieved when the number of modules is set to 4.

