# OpenReview forum: "Unsupervised Reinforcement Learning by Maximizing Skill Density Deviation"
_ICLR.cc/2025/Conference — Submitted to ICLR 2025_

### Official Review · Reviewer_ufVf · 2024-10-26

**Soundness:** 3
**Presentation:** 3
**Contribution:** 3
**Rating:** 5
**Confidence:** 3

**Summary:**

This paper addresses the challenge of discovering meaningful skills in large-scale spaces. The authors propose a novel skill discovery objective that maximizes state density deviation for each skill, introducing a CVAE with soft modularization. Additionally, to promote intra-skill exploration, they provide an intrinsic reward with theoretical proof in tabular MDP settings.

**Strengths:**

- The overall framework architecture, which incorporates a soft modularization technique to maximize state-density deviation in skill discovery for large-scale observations, is novel.
- The authors propose effective solutions to address both inter-skill state diversity and intra-skill exploration.
- The authors conduct comprehensive experiments across various algorithms and environments, including both state-based and pixel-based observations.
- Ablation studies further highlight the design choices of SD3.

**Weaknesses:**

- It would be beneficial for the authors to visualize the activations of different networks for various skills, especially in state-based observations. In such environments, SD3 may potentially use similar networks for different skills, as the observation spaces are relatively low-dimensional.
- It would be beneficial for the authors to visualize skill discovery in SD3 for both state-based and pixel-based environments, as shown on the left side of Figure 2.
- Is the same network size used for both SD3 and the baseline algorithms? I wonder if SD3, which utilizes soft modularization with a CVAE architecture, requires a larger network size. If so, the authors should provide additional experiments using the same network parameters for the baselines.

**Questions:**

- The authors should specify which environments were used for each experiment. Were the robustness and ablation experiments conducted under pixel-based observation settings?
- SD3 considers a discrete skill space Z, and the number of skills is a hayperparameter. Is this setting identical for all other algorithms?
- I am willing to increase my score, if the authors address the questions above.

---

> ### Author Response · Authors · 2024-11-24
> **Response to Reviewer ufVf**
>
> Dear Reviewer ufVf,
>
> We sincerely appreciate your precious time and constructive comments. In the following, we would like to answer your concerns separately.
>
> **W1**: Visualize the activations of different networks for various skills.
>
> **Response**: Thanks for the comment. We record the weights output by the routing network at the 1.5M step in pretraining phase and visualized them in the anonymized [link](https://anonymous.4open.science/r/visual_soft_module-8D2F/visual.pdf). In the visualization, the weights are depicted as connections between different modules, with dark red representing high probabilities and light red indicating low probabilities. It can be observed that some skills indeed share relatively similar network weights. However, as we only implemented two layers of modules, there remain differences in network configurations among different skills.
>
> **W2**: Visualize skill discovery in SD3 for both state-based and pixel-based environments.
>
> **Response**: Thanks for the comment. For both state-based and pixel-based environments, the process of skill discovery in SD3 remains consistent, as illustrated on the left side of Figure 2. The primary difference between state-based and pixel-based environments lies in the form of the input state. In the state-based environment, the state is processed sensor data, whereas in the pixel-based environment, the state consists of raw pixel data.
>
> **W3**: Provide additional experiments using the same network parameters for the baselines.
>
> **Response**: Thanks for the comment. After employing the soft-modularized CVAE, the model's parameter size has indeed increased due to the presence of multiple modules in each layer. Thus, we conduct experiments without the soft-modularized CVAE, using a standard CVAE where the encoder consists of a 4-layer MLP network. The results, shown in the table below, demonstrate that even without the soft-modularized CVAE, our method still achieves competitive performance on several downstream tasks.
>
> | Task              | CIC          | BeCL         | SD3          | SD3(w/o soft_modu) |
> | ----------------- | ------------ | ------------ | ------------ | ------------------ |
> | Quad Stand        | 700 $\pm$ 55 | 875 $\pm$ 33 | 847 $\pm$ 17 | 752 $\pm$ 64       |
> | Quad Walk         | 621 $\pm$ 69 | 743 $\pm$ 68 | 752 $\pm$ 40 | 642 $\pm$ 80       |
> | Quad Run          | 445 $\pm$ 36 | 535 $\pm$ 13 | 471 $\pm$ 13 | 422 $\pm$ 34       |
> | Quad Jump         | 565 $\pm$ 44 | 727 $\pm$ 15 | 676 $\pm$ 29 | 589 $\pm$ 45       |
> | Walker Stand      | 959 $\pm$ 2  | 952 $\pm$ 2  | 930 $\pm$ 5  | 910 $\pm$ 16       |
> | Walker Walk       | 903 $\pm$ 21 | 883 $\pm$ 34 | 914 $\pm$ 11 | 870 $\pm$ 30       |
> | Walker Run        | 450 $\pm$ 19 | 387 $\pm$ 22 | 451 $\pm$ 23 | 409 $\pm$ 55       |
> | Walker Flip       | 641 $\pm$ 26 | 611 $\pm$ 18 | 595 $\pm$ 25 | 523 $\pm$ 33       |
> | Jaco Top Left     | 149 $\pm$ 10 | 125 $\pm$ 10 | 142 $\pm$ 7  | 125 $\pm$ 5        |
> | Jaco Top Right    | 163 $\pm$ 9  | 126 $\pm$ 10 | 152 $\pm$ 7  | 117 $\pm$ 5        |
> | Jaco Bottom Left  | 154 $\pm$ 6  | 148 $\pm$ 13 | 151 $\pm$ 7  | 134 $\pm$ 8        |
> | Jaco Bottom Right | 149 $\pm$ 4  | 139 $\pm$ 14 | 152 $\pm$ 9  | 122 $\pm$ 8        |
>
> **Q1**: Were the robustness and ablation experiments conducted under pixel-based observation settings?
>
> **Response**: Thanks for the question. Our robustness and ablation experiments are conducted in state-based environments because the pixel-based experiments use Dreamer as the backbone, which inherently exhibits a certain degree of robustness to environmental perturbations.
>
> **Q2**: SD3 considers a discrete skill space Z, and the number of skills is a hayperparameter. Is this setting identical for all other algorithms?
>
> **Response**: Thanks for the question. The number of skills of all algorithms is set to 16.

---

### Official Review · Reviewer_GZJz · 2024-11-01

**Soundness:** 3
**Presentation:** 2
**Contribution:** 2
**Rating:** 5
**Confidence:** 2

**Summary:**

This work proposes a novel unsupervised reinforcement learning (RL) framework called SD3, aimed at improving skill discovery by maximizing state density deviation across skills. Unlike traditional entropy-based or Mutual Information (MI) techniques that often struggle in large state spaces, SD3 uses a conditional autoencoder with soft modularization to estimate skill-specific state densities, enabling robust and scalable skill discovery. It incorporates an intrinsic reward resembling count-based exploration in a latent space to encourage inter-skill diversity and intra-skill exploration. Extensive experiments demonstrate that SD3 yields diverse, meaningful skills significantly enhancing performance across various downstream tasks.

**Strengths:**

1. The paper is well-written and effectively compares the proposed method with baselines, providing a thorough analysis that includes detailed mathematical proofs.
2. The paper employs soft modularization for better estimation of the state density across all skills, which is known to be challenging to estimate.
3. The paper performs ablation studies to assess the influence of various design choices made in this research.

**Weaknesses:**

1. The exploration reward represented by the KL term does not guarantee that it always reaches the ‘outer’ region (i.e., frontier states) of ‘all’ skills, implying it may not serve as an optimal reward for exploration. Additionally, since the CVAE encoder is trained simultaneously with RL, this effect could be even more pronounced.
2. Choreographer[1] uses VAE and a KL term reward for exploration. While they use a world model for skill learning, it is quite similar to the proposed method. The authors should compare and explain the differences.
3. The overall algorithm is limited to a discrete skill space due to the architectural design of soft modularization.
4. Minor Comment: The visualization of soft modularization in Figure(1)-a could be improved to make it more intuitive and easier to understand at a glance.

[1] Mazzaglia, Pietro, et al. "Choreographer: Learning and adapting skills in imagination." arXiv preprint arXiv:2211.13350 (2022).

**Questions:**

1. In the state-based URLB experiments (Figure 4), I have some questions regarding the comparison with baselines:
    1. When the baseline algorithm allows for both continuous and discrete skill spaces, which one did you choose?
    2. How did you determine the dimensionality of the skill space?
    3. Additionally, did you perform fine-tuning over all skills or did you select a specific skill $z^*$ that performed best on the downstream task before fine-tuning?

    For instance, in the case of METRA[1], both discrete and continuous skills are possible, and it's also possible to obtain $z^*$ in a zero-shot manner. I’m curious if this was taken into consideration.

2. I'm curious why BECL[2] was not included in the comparison for the tree-like maze in Figure 6. According to the BECL report, it appears to explore a broad region in the same tree-like maze and shows competitive skill distinguishability. Given that BECL also optimizes skill distinguishability using a contrastive style while considering exploration, it seems appropriate to include it in the comparison.


[1] Park, Seohong, Oleh Rybkin, and Sergey Levine. "Metra: Scalable unsupervised rl with metric-aware abstraction." arXiv preprint arXiv:2310.08887 (2023).

[2] Yang, Rushuai, et al. "Behavior contrastive learning for unsupervised skill discovery." International Conference on Machine Learning. PMLR, 2023.

---

> ### Author Response · Authors · 2024-11-24
> **Response to Reviewer GZJz(Part 1/2)**
>
> Dear Reviewer GZJz,
>
> We sincerely appreciate your precious time and constructive comments. In the following, we would like to answer your concerns separately.
>
> **W1**: The exploration reward represented by the KL term.
>
> **Response**: Thanks for the comment and we want to clarify the following points:
>
> (1) In Section 3.3 and the Appendix A.3, we have demonstrated that the KL term, when used as an exploration bonus, resembles a generalized UCB-bonus and is provably efficient in linear MDPs. And then, we validate the effectiveness of this reward in encouraging exploration through experimental results.
>
> (2) Additionally, using the KL term as an exploration bonus is not uncommon. For example, LBS[1] also employs a KL-based intrinsic reward design, as shown in Equation (2) of their paper. Similar to our approach, LBS maps states to latent variables $z$ and uses the KL divergence between the prior and approximate posterior distributions of $z$ to quantify the agent's uncertainty about a transition. This KL divergence is then used as an intrinsic reward to encourage exploration. Furthermore, LBS also demonstrates the effectiveness of this intrinsic reward design in exploration through state coverage experiments.
>
> (3) Training the CVAE jointly with RL agent does not adversely affect the exploration reward. This is because the CVAE is trained on data sampled from the replay buffer. As training progresses, the CVAE becomes increasingly accurate at reconstructing frequently visited states. Consequently, the KL divergence between the prior and approximate posterior distributions of the latent variable $h$ diminishes for such states, reducing the exploration reward. Conversely, states that are less frequently visited yield a larger KL divergence, thus providing the agent with a higher exploration reward.
>
> [1] Mazzaglia, Pietro, et al. "Curiosity-driven exploration via latent bayesian surprise." Proceedings of the AAAI conference on artificial intelligence.
>
> **W2**: Choreographer also uses VAE and a KL term reward for exploration.
>
> **Response**: Thanks for the comment. Although Choreographer also utilizes VAE and KL term, its approach is fundamentally different from the method we propose. The differences are primarily reflected in the following two aspects:
>
> (1) **Use of VAE.** SD3 integrates skill discovery and skill learning into a unified process. We use a soft modularized CVAE to estimate the state density $d^{\pi}_z(s)$ corresponding to each skill. By maximizing the deviation of state densities between different skills, we simultaneously promote skill discovery and skill learning. In contrast, Choreographer separates the processes of skill discovery and skill learning, treating skill discovery as a form of state representation learning. Specifically, it employs VQ-VAE to learn a codebook for states and then considers the codes in the codebook as skills in the environment.
>
> (2) **Use of KL Term.** In SD3, the KL term is used to calculate the divergence between the posterior distribution of the latent variable $h$ conditioned on $(s,z)$ and the posterior distribution of $h$. This serves as a measure of skill-conditioned state novelty, encouraging state exploration within a skill. On the other hand, Choreographer employs KL terms in two distinct contexts: (a) during world model training, to measure the divergence between the model's posterior and prior, and (b) in scenarios without pre-collected data, where the same KL term used for training the world model is repurposed as an intrinsic reward for learning an exploration actor-critic.
>
> Meanwhile, although Choreographer also conducts experiments on state-based URLB, its reported results are based on experiments performed on pre-collected datasets of exploratory data. This is inconsistent with our fully online experimental setting, making direct comparisons with its reported performance inappropriate. Additionally, while Choreographer’s open-source codebase provides a parallel exploration version, it simultaneously trains a world model during the skill learning phase. Since the world model can summarize past experiences to predict future states, it significantly enhances the learned policy. Consequently, comparing such an approach with our model-free method is also unfair and inappropriate.

---

> ### Author Response · Authors · 2024-11-24
> **Response to Reviewer GZJz(Part 2/2)**
>
> **W3**: The overall algorithm is limited to a discrete skill space due to the architectural design of soft modularization.
>
> **Response**: Thanks for the comment. In response to the concern that SD3 can only discover discrete skills, it's important to note that while continuous skill spaces may seem advantageous, they do not necessarily guarantee better performance than discrete ones. As demonstrated in Figure 3 of our paper, methods like DADS, which utilize a continuous skill space, still show limited performance. Similarly, in the URLB experiment, baseline methods such as APS and CIC also work on continuous skill spaces, yet they do not lead to superior outperformance. Although learning an infinite number of diverse and meaningful skills is theoretically desirable, it remains a challenge for current skill discovery methods.
>
> In SD3, our optimization objective for a specific $(s,z)$ is formulated as $I_{\text{SD3}}(s,z)\triangleq\log\frac{\lambda d_z^\pi(s)}{\lambda d_z^\pi(s)p(z)+\sum_{z'\neq z}d_{z'}^\pi(s)p(z')}$. We denote the state density of other skills $\{z^{\prime}\}$ except for $z$ as $\rho_{z^{c}}\triangleq\sum_{z^{\prime}\neq z}d_{z^{\prime}}^{\pi}(s)$. It can be observed that in a continuous skill space (infinite skills), accurately estimating the value of $\rho_{z^{c}}$ becomes challenging. This difficulty can hinder the effective promotion of deviations in exploration regions between different skills, thereby compromising the diversity of the learned skills. This is indeed a limitation of our method, and we have addressed it in the conclusion.
>
> **W4**: Minor Comment: The visualization of soft modularization in Figure(1)-a could be improved to make it more intuitive and easier to understand at a glance.
>
> **Response**: Thanks for the comment and we will improve the Figure(1)-a in the revised version.
>
> **Q1**: Questions regarding the comparison with baselines.
>
> **Response**: Thanks for the question and we have addressed each of the issues you raised as follows:
>
> (1) For a fair comparison, we chose a discrete skill space for methods like METRA, which allow for both continuous and discrete skill spaces.
>
> (2) Following the standard in URLB, we set the skill dimension to 16 for all methods.
>
> (3) Regarding this issue, we have provided a detailed discussion in Appendix D.5 and comparison in Figure 12. In the main experiments of the paper, for a fair comparison, we adhered to the standards set in the URLB, employing a random sampling skills method during the fine-tuning stage to evaluate the average performance of skills. However, in Appendix D.5, we additionally discussed two alternative methods for skill selection: *regress-meta* and *meta-controller*. Regress-meta computes the expected reward for each skill during the first 4K steps of the fine-tuning phase to determine its skill-value, and then selects the skill with the highest skill-value to perform the downstream task. The meta controller, on the other hand, involves training a high-level controller $\mu(z|s)$ during fine-tuning phase to select the most appropriate skill for the current state $s$.
>
> We conducted experiments with both skill selection methods, and the results are shown in Figure 12. Using regress-meta for skill selection yields a slight performance improvement over random skill selection. The meta-controller, however, demonstrated poorer performance. We hypothesized that the meta-controller usually requires a large number of examples to train, which is difficult to converge within the 100K fine-tuning steps.
>
> Additionally, the strategy used in METRA to obtain $z^*$ is not applicable to our case, as it requires computing based on the goal state of the downstream task. However, in our experiment environment, there is no explicitly defined goal state.
>
> **Q2**: Why BECL was not included in the comparison for the tree-like maze?
>
> **Response**: Thanks for the question. Our tree-like maze experiment is primarily designed to visualize that SD3 is still able to learn distinguishable skills and explore distant regions of the environment as the number of skills increases. DIAYN and DADS are included as comparisons to illustrate scenarios where skills are distinguishable but the explored regions are limited as the number of skills increases, thereby highlighting SD3's effective exploration capabilities. While BeCL also performs well in the tree-like maze, the limited spatial extent of the environment constrains its ability to provide meaningful visual insights into the comparative performance of the two methods.

---

> ### Comment · Reviewer_GZJz · 2024-12-03
>
> Thanks to the author for a detailed response to my questions and for providing additional insights. I don't have any further questions.

---

### Official Review · Reviewer_tRjS · 2024-11-04

**Soundness:** 2
**Presentation:** 3
**Contribution:** 2
**Rating:** 5
**Confidence:** 5

**Summary:**

The paper presents a novel unsupervised reinforcement learning (URL) approach named State Density Deviation of Different Skills (SD3).
SD3 maximizes skill density deviation to make the skill distinguishable and utilizes the KL divergence between the posterior distribution and the prior distribution of the latent variable as the exploration bonus for state coverage. To estimate skill density $d_z^\pi(s) = p(s|z)$, SD3 adopts a routing-network-based conditional variational autoencoder to estimate its lower bound.
SD3 is evaluated against a range of existing URL baselines on both state-and pixel-based URL Benchmark (URLB) to show it effectiveness.

**Strengths:**

In general, this paper is well-written and easy to follow. The experiments are extensive, involving multiple baseline methods across both state- and pixel-based URL environments. Especially,

**The skill-density perspective of mutual information is interesting.**

As one of the main contributions, SD3 presents a new perspective of mutual information, that is,

$$I(S;Z) = \mathbb{E}\_{z,s} [\log\frac{ p(s|z)}{p(s)}] = \mathbb{E}\_{z,s} [\log\frac{ p(s|z)}{p(s|z)p(z) + \sum\_{z'\neq z}p(s|z')p(z')}].$$

By calling $p(s|z):=d\_z^\pi(s)$ the "skill density", SD3 proposes a soft mutual information as

$$I_{\text{SD3}}(S;Z) = \mathbb{E}\_{z,s} [\log\frac{ {\color{red}\lambda}p(s|z)}{{\color{red}\lambda}p(s|z)p(z) + \sum\_{z'\neq z}p(s|z')p(z')}.$$

**Introducing the routing network to model the skill-conditioned state encoder is novel.**

Introducing the routing network to model the skill-conditioned state encoder is novel for me. However, I have some concerns about the adoption of this architecture, which was originally designed for a multi-task policy network. I will discuss this in the weaknesses part.

**Weaknesses:**

The paper presents interesting contributions but suffers from several key weaknesses that diminish its impact.

**W1: The motivation for introducing ${\color{red}\lambda}$ in $I_{\text{SD3}}(S;Z)$ is unclear and lacks theoretical grounding.**

While it’s established that $\max I(S; Z) = \max I_{\text{SD3}}(S; Z) = \max H(Z)$, indicating that maximizing mutual information between $S$ and $Z$ implies $H(S|Z) = H(Z|S) = 0$, the purpose of incorporating ${\color{red}\lambda}$ into the optimization objective remains ambiguous. The authors state (Line 145) that “increasing ${\color{red}\lambda}$ will weaken the gradient of SD3, reducing the state densities of other skills and preventing skill collapse in SD3.” However, the link between reducing state densities and preventing skill collapse is not self-evident and warrants further explanation.

**W2: Extending the intra-skill exploration bonus $r_z^{\text{exp}}$ to continuous state spaces is problematic.**

With $r(h)$ set as a standard Gaussian $\mathcal{N}(\textbf{0}; \textbf{1})$ (Line 244), $r_z^{\text{exp}} = D_{\text{KL}}[Q(h|s,z) || r(h)]$ represents the KL divergence between Gaussian distributions since $h = \mu(s,z) + \sigma(s,z) * \epsilon$ is a Gaussian distribution (Appendix B.2, Line 1059). This allows for an analytic expression of $r_z^{\text{exp}}$:

$$
r_z^{\text{exp}} = \frac{1}{2} \{\mu^{T}\mu + \text{tr}\{\sigma\} - k - \log|\sigma|\}.
$$

However, this form of $r_z^{\text{exp}}$ lacks a clear connection to state novelty or surprise, a critical component of exploration. Established methods encourage exploration by approximating state novelty with pseudo-counts $r \approx 1/p(s)$ or information content $r \approx -\log p(s)$. In contrast, $r_z^{\text{exp}} = D_{\text{KL}}[Q(h|s,z) || r(h)]$ does not exhibit such properties, making it unclear how it incentivizes novel state visits. Although Theorem 3.2 implies that $r_z^{\text{exp}}$ resembles a count-based exploration bonus in tabular MDPs, its role in continuous cases requires further clarification. Since all the experiments are carried out in continuous cases, providing the analysis only in simple tabular cases is not convincing.

By the way, there is a typo on Line 238, where $D_{\text{KL}}[Q(h|s,z) || P(h)]$ should be referenced as the upper bound of $I(S; H|Z) = D_{\text{KL}}[Q(h|s,z) || P(h|z)]$.

**W3: Lack of a detailed description of the routing-based CVAE’s critical module.**

Although Figure 1 outlines the high-level architecture, a detailed description of the core module in the routing-based CVAE would enhance clarity. This additional explanation would help convey the structural intricacies that underpin the architecture.

**W4: Experiments in the Maze environment are misleading.**

The Maze environment is primarily a visualization tool for 2D skills, yet SD3 only uses CIC as a baseline here. Given that SOTA methods such as BeCL and CeSD outperform SD3 in this setting, omitting these baselines could mislead readers about SD3's comparative performance. BeCL’s results in Figure 4, comparable to CIC in state-based URLB, further highlight the need for a fairer baseline selection.

**W5: Missing SOTA baselines in state-based URLB.**

Although the authors include SMM, DIAYN, ICM, APS, Disagreement, CSD, RND, Metra, ProtoRL, and APT baselines, these approaches are not sufficiently competitive. More recent SOTA methods like MOSS (NeurIPS'22), EUCLID (ICLR'22), and CeSD (ICML'24) should also be considered. Moreover, as demonstrated in the paper, the performance gains of SD3 over CIC—a weaker baseline than MOSS, BeCL, and CeSD—are marginal. Thus, the current baselines in both the Maze and URLB settings fail to support SD3’s effectiveness convincingly.

**W6: Robustness experiments lack rigor.**

A key claim of SD3 is that $r_z^{\text{exp}}$ offers more robustness than the entropy-based bonus $r = -\log p(s)$. However, as noted in W2, the relationship between $r_z^{\text{exp}}$ and a pseudo-count bonus $r \approx 1/p(s)$ in continuous state spaces remains unclear. Additionally, the robustness evaluation design is unconventional; measuring performance on downstream tasks does not adequately reflect the robustness of the policy network. A more effective approach would involve injecting noise during the fine-tuning phase to assess resilience against adversarial perturbations. This design aligns with robustness assessments in adversarial learning, where robustness-accuracy trade-offs are common. Finally, robustness might also depend on the model’s parameter count, so a parameter comparison between SD3 and CIC is essential for fair comparison.

1. **[MOSS]** Zhao, Lin, Li, Liu, & Huang. *A Mixture Of Surprises for Unsupervised Reinforcement Learning.* NeurIPS, 2023.
2. **[CeSD]** Bai, Yang, Zhang, Xu, Chen, Xiao, & Li. *Constrained Ensemble Exploration for Unsupervised Skill Discovery.* ICML, 2024.

**Questions:**

Please see the weakness part for detailed questions.

---

> ### Author Response · Authors · 2024-11-24
> **Response to Reviewer tRjS(Part 1/3)**
>
> Dear Reviewer tRjS,
>
> We sincerely appreciate your precious time and constructive comments. In the following, we would like to answer your concerns separately.
>
> **W1**: The motivation for introducing $\lambda$ in $I_{\mathrm{SD3}}(S;Z)$ is unclear and lacks theoretical grounding.
>
> **Response**: Thanks for the comment and we want to clarify the following points:
>
> (1) First, we want to clarify that $\max I(S;Z)=\max I_{\mathrm{SD3}}(S;Z)$ holds if and only if $\lambda=1$, as proven in Section 3.1 and the Appendix A.2 of the paper. Furthermore, increasing $\lambda$ does not directly reduce the state densities of other skills but rather slows down the rate at which the state densities of other skills decrease.
>
> (2) Second, skill collapse refers to an extreme scenario that may occur when optimizing $I_{SD3}$, where each skill tends to visit a distinct state that other skills do not access. This leads to limited state coverage for each skill. To address this issue, one approach is to slow down the rate at which the state densities of other skills decrease. When we compute the gradient of the state density of other skills, it can be expressed as $\nabla_{\rho_zc}I_{\mathrm{SD3}}(s,z)=-1/(\lambda d_z^\pi(s)+\rho_{z^c}(s))$. As $\lambda$ increases, the gradient responsible for reducing the state densities of other skills becomes smaller, thereby mitigating the issue.
>
> **W2**: Extending the intra-skill exploration bonus $r_z^{exp}$ to continuous state spaces is problematic.
>
> **Response**: Thanks for the comment and we want to clarify the following points:
>
> (1) In the CVAE, $\mu$ and $\sigma$ represent the mean vector and covariance matrix of the approximate posterior distribution $q(h|s, z)$, respectively, where $k$ is the dimensionality of the latent space $Z$. Therefore, the terms in $\frac12\mu^T\mu+\text{tr}\sigma-k-\log|\sigma|$ have the following meanings:
> * $\mu^{T}\mu$ quantifies the deviation of the posterior mean from the standard Gaussian prior (with mean zero). For frequently visited states, the encoder’s output mean $\mu$ tends to approach zero, indicating that the posterior aligns closely with the prior. Conversely, for novel states, $\mu$ deviates significantly from zero, resulting in a larger $\mu^{T}\mu$.
> * $\mathrm{tr}(\sigma)$ reflects the encoder’s uncertainty about the state. For frequently visited states, $\sigma$ tends to approach the identity matrix $I$, whereas for novel states, the encoding uncertainty increases, leading to a higher $\mathrm{tr}(\sigma)$.
> * $-\log(|\sigma|)$ represents the negative entropy of the posterior distribution. For frequently visited states, lower uncertainty causes $-\log(|\sigma|)$ to increase, while for novel states, higher uncertainty results in a decrease in $-\log(|\sigma|)$.
>
> When analyzing these components for novel states individually, the first two terms ($\mu^{T}\mu$ and $\mathrm{tr}(\sigma)$) increase, reflecting higher deviation and uncertainty. Meanwhile, the third term ($-\log(|\sigma|)$) decreases as a result of the higher entropy associated with novel states. However, since $\mathrm{tr}(\sigma) - \log(|\sigma|)$ can be re-expressed as $\sum (\sigma_i - \log(\sigma_i))$, where $\sigma_i$ represents the diagonal elements of the covariance matrix, this term remains strictly positive and grows as $\sigma_i$ deviates further from $1$. Consequently, for novel states, the combined term $\mathrm{tr}(\sigma) - \log(|\sigma|)$ increases, leading to an overall increase of $r_z^{\exp}=\frac12\mu^T\mu+\mathrm{tr}\sigma-k-\log|\sigma|$.
>
> (2) Additionally, in Appendix A.3, our proof is not restricted to tabular MDPs but is first established for linear MDPs. Under the linear cases, $r^{\text{exp}}$ resembles a generalized UCB-bonus (i.e.,$\eta_t^\top\Lambda_t^{-1}\eta_t$, see Equation (21) in our paper). Based on this, we further derive that in tabular cases, $r^{\exp}\approx\frac{|\mathcal{S}|/2}{N(s,z)+\kappa}$. Accordingly, we would like to elaborate on two key aspects:
> * The UCB-term in exploration has been proven to be provable efficient in linear MDPs, which has been rigorously studied in previous research (e.g., Alg. 1 and Theorem 3.1 in reward-based settings [1], and Alg. 2 and Theorem 4.2 in reward-free settings [2]).
> * The linear MDP assumption is widely adopted in the theoretical study of reinforcement learning with function approximation [3,4]. From a technical perspective, this linear setting can naturally be extended to the (infinite-dimensional) kernelized linear setting or the neural network setting by leveraging the neural tangent kernel regime [5].
>
> Based on these two points, we argue that the proposed $r^{\text{exp}}$ is well-suited for continuous cases.

---

> ### Author Response · Authors · 2024-11-24
> **Response to Reviewer tRjS(Part 2/3)**
>
> (3) We will correct the typo you mentioned in the revised version.
>
> [1] Jin C, Yang Z, Wang Z, et al. Provably efficient reinforcement learning with linear function approximation. Conference on learning theory, 2020
>
> [2] Zhang W, Zhou D, Gu Q. Reward-free model-based reinforcement learning with linear function approximation. NeurIPS, 2021
>
> [3] Chi Jin, Zhuoran Yang, Zhaoran Wang, and Michael I. Jordan. Provably Efficient Reinforcement Learning with Linear Function Approximation. In COLT. 2020
>
> [4] Ying Jin, Zhuoran Yang, and Zhaoran Wang. Is Pessimism Provably Efficient for Offline RL? In ICML 2021.
>
> [5] Zhuoran Yang, Chi Jin, Zhaoran Wang, Mengdi Wang, and Michael I. Jordan. On Function Approximation in Reinforcement Learning: Optimism in the Face of Large State Spaces. In NeurIPS 2020.
>
> **W3**: Lack of a detailed description of the routing-based CVAE’s critical module.
>
> **Response**: Thanks for the comment. In the soft-modularized CVAE, both the encoder and decoder basic networks consist of two module layers, with each module layer containing four modules. And then each module is composed of a Batch Normalization layer followed by an MLP layer.
>
> **W4**: Experiments in the Maze environment are misleading.
>
> **Response**: Thanks for the comment. The maze experiment is primarily designed to visualize tha SD3 can discover distinguishable skills while effectively exploring the environment. In this process, CIC is used as a baseline to illustrate a scenario where exploration is strong, but the discovered skills are highly entangled. DIAYN and DADS are included as baselines to represent scenarios where the skills are distinguishable, but the exploration is limited to smaller regions of the environment. While BeCL and CeSD also perform well in the maze, the limited spatial extent of the maze environment constrains their ability to provide meaningful visual insights into the performance differences among the methods.
>
> **W5**: Missing SOTA baselines in state-based URLB.
>
> **Response**: Thanks for the comment. EUCLID is a model-based method, whereas SD3 and the baselines discussed in this paper are all model-free methods, making a direct comparison with EUCLID inappropriate. Regarding MOSS and CeSD, their reported performance is compared to SD3 as shown in the table below.
>
> | Task              | MOSS         | CeSD         | SD3          |
> | ----------------- | ------------ | ------------ | ------------ |
> | Quad Stand        | 911 $\pm$ 11 | 919 $\pm$ 11 | 847 $\pm$ 17 |
> | Quad Walk         | 635 $\pm$ 36 | 889 $\pm$ 23 | 752 $\pm$ 40 |
> | Quad Run          | 485 $\pm$ 6 | 586 $\pm$ 25 | 471 $\pm$ 13 |
> | Quad Jump         | 674 $\pm$ 11 | 755 $\pm$ 14 | 676 $\pm$ 29 |
> | Walker Stand      | 962 $\pm$ 3  | 960 $\pm$ 3  | 930 $\pm$ 5  |
> | Walker Walk       | 942 $\pm$ 5 | 834 $\pm$ 34 | 914 $\pm$ 11 |
> | Walker Run        | 531 $\pm$ 20 | 337 $\pm$ 19 | 451 $\pm$ 23 |
> | Walker Flip       | 729 $\pm$ 40 | 541 $\pm$ 17 | 595 $\pm$ 25 |
> | Jaco Top Left     | 150 $\pm$ 5 | 215 $\pm$ 4 | 142 $\pm$ 7  |
> | Jaco Top Right    | 150 $\pm$ 6  | 195 $\pm$ 9 | 152 $\pm$ 7  |
> | Jaco Bottom Left  | 151 $\pm$ 5  | 208 $\pm$ 5 | 151 $\pm$ 7  |
> | Jaco Bottom Right | 150 $\pm$ 5  | 186 $\pm$ 13 | 152 $\pm$ 9  |
>
> It can be observed that, apart from the Jaco Arm domain where CeSD demonstrates significant advantages, SD3 remains competitive across other downstream tasks. Additionally, to further illustrate the strengths of SD3, we conduct an additional experiment to compare the robustness of CeSD and our method by introducing Gaussian noise into the state space. In light of the weakness 6 you pointed out, we add noise to the environment during both the pretraining and fine-tuning phases and compare the results using the SD3 based on a standard CVAE structure. The results are shown in the table below.
>
>
> | Task       | CeSD (Noisy) | CeSD (Normal) | Performance Ratio | SD3(Noisy)   | SD3(Normal)  | Performance Ratio |
> | ---------- | ------------ | ------------- | ----------------- | ------------ | ------------ | ----------------- |
> | Quad Stand | 511 $\pm$ 77 | 919 $\pm$ 11  | **55.60%**        | 561 $\pm$ 64 | 752 $\pm$ 64 | **74.60%**        |
> | Quad Walk  | 294 $\pm$ 27 | 889 $\pm$ 23  | **33.07%**        | 405 $\pm$ 45 | 642 $\pm$ 80 | **63.08%**        |
> | Quad Run   | 408 $\pm$ 21 | 586 $\pm$ 25  | **69.62%**        | 381 $\pm$ 27 | 422 $\pm$ 34 | **90.28%**        |
> | Quad Jump  | 306 $\pm$ 38 | 755 $\pm$ 14  | **40.53%**        | 419 $\pm$ 45 | 589 $\pm$ 45 | **71.14%**        |
> | Average    | --           | --            | **49.71%**        | --           | --           | **74.77%**        |
>
> The result shows that although CeSD outperforms our method in a standard URLB environment, our method significantly outperforms CeSD in a noisy setting, which demonstrates our unique theoretical advantages and technique contributions.

---

> ### Author Response · Authors · 2024-11-24
> **Response to Reviewer tRjS(Part 3/3)**
>
> **W6**: Robustness experiments lack rigor.
>
> **Response**: Thanks for the comment. As explained in our response to W2, in linear cases, the proposed exploration reward $r^{exp}$ resembles a generalized UCB-bonus and is provably efficient under linear settings.
>
> Additionally, in our response to W5, we follow your suggestion and introduce noise into the environment during the fine-tuning phase as well. The results of robustness comparison with CIC is presented in the table below.
>
> | Task       | CIC (Noisy) | CIC (Normal) | Performance Ratio | SD3(Noisy)   | SD3(Normal)  | Performance Ratio |
> | ---------- | ------------ | ------------- | ----------------- | ------------ | ------------ | ----------------- |
> | Quad Stand | 392 $\pm$ 45 | 700 $\pm$ 55  | **56.00%**        | 561 $\pm$ 64 | 752 $\pm$ 64 | **74.60%**        |
> | Quad Walk  | 209 $\pm$ 16 | 621 $\pm$ 69  | **33.67%**        | 405 $\pm$ 45 | 642 $\pm$ 80 | **63.08%**        |
> | Quad Run   | 201 $\pm$ 51 | 445 $\pm$ 36  | **45.17%**        | 381 $\pm$ 27 | 422 $\pm$ 34 | **90.28%**        |
> | Quad Jump  | 322 $\pm$ 30 | 565 $\pm$ 44  | **56.99%**        | 419 $\pm$ 45 | 589 $\pm$ 45 | **71.14%**        |
> | Average    | --           | --            | **47.96%**        | --           | --           | **74.77%**        |
>
> It can be observed that the robustness of our method remains superior to that of CIC under this experiment setting.

---

> > ### Comment · Reviewer_tRjS · 2024-11-25
> > **Reply to Author Response**
> >
> > Thanks to the authors for their detailed response. The insight provided by the authors for **W2** is interesting. I now understand the relationship between the intra-skill exploration bonus $r^\text{exp}_z$ and the novelty-based intrinsic bonus. However, for other weaknesses, I cannot be thoroughly convinced. Below are my remaining concerns.
> >
> > **W1 Motivation for introducing \(\lambda\):**
> >
> > Although the authors provide insight for introducing \(\lambda\), i.e., \(\lambda\) reduces the gradient responsible for decreasing state densities of other skills, such a claim needs a clearer investigation. For instance, how do such gradients vary when \(\lambda\) changes?
> >
> > By the way, regarding \(I_{\text{SD3}}\), it is actually a "soft" mutual information \(I\) (another interpretation of mutual information). Thus, I believe that it will be clearer that the authors utilize "density deviation" as an insight for \(I_{\text{SD3}}\) instead of renaming such "soft" mutual information as "density deviation."
> >
> > **W4 Experiments in the Maze environment are misleading:**
> >
> > I still believe comparing SD3 with CIC instead of CeSD is unfair and misleading. Now that "the limited spatial extent of the maze environment constrains their ability to provide meaningful visual insights into the performance differences among the methods," it is the authors' responsibility to design an appropriately complex maze environment to evaluate SD3 and CeSD instead of simply avoiding mentioning CeSD.
> >
> > **W5 Performance in state-based URLB compared with SOTA baselines:**
> >
> > From the latest table provided by the authors, SOTA baselines, including MOSS and CeSD, outperform SD3 in most subtasks. This result weakens the effectiveness of SD3. Although marginal improvement is not always necessary for a good algorithm, in the current version of this paper, the robustness analysis of SD3 is more likely to be an "accessory" to amend the marginal improvement of SD3 in performance.
> >
> > **W6 Robustness of SD3:**
> >
> > Although the authors show that SD3 outperforms CIC in a noisy environment, the reason behind such robustness is unclear. As I mentioned before, the robustness can stem from many facts. Since SD3 includes many components, the contribution of each element to the robustness needs to be carefully investigated.
> >
> > Based on the above concerns, I will keep a cautious attitude and improve my score to 5. In sum, although SD3 seems promising to improve robustness, this paper needs major revision and re-organization to further investigate the reason behind robustness and make the presentation clearer.

---

### Official Review · Reviewer_z324 · 2024-11-07

**Soundness:** 3
**Presentation:** 3
**Contribution:** 2
**Rating:** 5
**Confidence:** 4

**Summary:**

This paper proposes an unsupervised reinforcement learning algorithm that maximizes inter-skill state visitation diversity and intra-skill exploration through a clean formulation. It provides solid theoretical analysis as well as good empirical experimental results on the unsupervised reinforcement learning benchmark under both state-based and image-based settings.

**Strengths:**

1. The presentation and writing of this paper is clear.
2. The proposed inter-skill state deviation and intra-skill exploration objectives are clean, with detailed and solid theoretical analysis.
3. The main experiments, visualization, and ablation studies demonstrate the effectiveness of the proposed method.

**Weaknesses:**

1. Experiments are not convincing enough.

    The method is mainly tested using standard URLB evaluation protocol, which can be questionable. DDPG is pre-trained with intrinsic rewards. Its value function will mismatch the fine-tuning distribution with extrinsic rewards. Also using "random selection" or "regress-meta" is not convincing enough to show if the agents can truly learn useful skills during pre-training, since the policy and value are *largely re-learned after the reward distribution change*. It couples the RL online sample efficiency with the skill discovery pre-training, which is not a good standalone metric for skill discovery.

    Instead, [1] uses a high-level controller that learns to select from **frozen skills** learned from pre-training, and also measures the total state coverage, which is a better metric for exploration and skill discovery during pre-training. [1] also provides more visualizations of the learned skills in broader domains, while this paper only showcased the maze examples. It could be more convincing if this paper could also achieve SoTA using a better metric but not limited to URLB's evaluation.

    [1] Park et al., METRA: Scalable Unsupervised RL with Metric-Aware Abstraction

2. Comparisons are not fair. The proposed method benefits from soft modularization, but the MI-based baselines do not use soft modularization. The performance of the proposed method *without* the soft modularization cVAE should be reported in the main experiments.
3. The method could be sensitive to hyperparameters, e.g. softmax temperature. From Figure 10 (a), it seems the temperature can have a large impact on the performance. Is that true if the temperature is close to 1 (the default choice)? What's the number of modules ($m$) in cVAE?

**Questions:**

1. Why use PPO for the maze visualization example, while DDPG is used as the main results in URLB?
2. Line 210, of length $l+1$ -> of layer $l+1$ , or of shape $m \times m$? Also softmax temperature is not mentioned in Line 212.
3. In the robustness experiment: "We conduct experiments in noisy domains of URLB by adding noise during pre-training". The noise is added on states, or transitions (states and actions)?
4. The results in Table 2 and Table 3 are partially highlighted, which is confusing.
5. What's the number of modules ($m$) in cVAE?  It's not shown in the appendix, and also ablations on the numbers are needed.

The reviewer is willing to adjust the score if the above questions and concerns are properly addressed.

---

> ### Author Response · Authors · 2024-11-24
> **Response to Reviewer z324(Part 1/3)**
>
> Dear Reviewer z324,
>
> We sincerely appreciate your precious time and constructive comments. In the following, we would like to answer your concerns separately.
>
> **W1**: Experiments are not convincing enough.
>
> **Response**: Thanks for the comment and we want to clarify the following points:
>
> (1) ULRB is a widely used benchmark in the field of unsupervised reinforcement learning (URL) to evaluate algorithm performance. Many skill discovery methods, such as CIC[1], BeCL[2], and Choreographer[3], have employed this benchmark for their experiments. Additionally, under the constraint of limited online samples, discovering and learning useful skills during pretraining, followed by fine-tuning these learned skills in downstream tasks, can significantly enhance the efficiency and performance of policy learning compared to learning from scratch. Although both the policy and value function are re-learned during the fine-tuning phase, the interaction time steps available for learning are very limited (only 100K steps). Consequently, the agent's performance in downstream tasks heavily relies on whether diverse behaviors and useful skills were acquired during pretraining. Therefore, it is appropriate to use this as an evaluation metric for skill discovery methods.
>
> (2) For the skill selection strategy, we have observed that METRA employs a high-level controller during downstream tasks to select skills from frozen skills learned during pretraining to maximize rewards. However, the authors of METRA did not provide the implementation details of this high-level controller or the downstream task execution methods in their codebase. As mentioned in Appendix E.5, we attempted to implement a high-level controller, but the performance did not reach the level achieved by METRA. This is likely because it is difficult to adequately train a high-level controller within a limited 100K-step fine-tuning phase. We are actively reaching out to the METRA authors to get their high-level controller implementation.
>
> (3) Based on your suggestion, we introduce an additional metric to assess the exploration capability of the method. Specifically, we use one skill to sample 10,000 states in the environment with the pretrained policy and utilize the particle entropy estimation from the APT method to calculate the mean Euclidean distance between each sampled state and its $k$-nearest neighbors (we set $k=6$), which serves as the entropy of the visited state. This entropy is then used as a metric of the method's exploration capability and the results are shown in the table below.
>
> | Method  | DIAYN            | METRA            | CIC              | BeCL             | SD3              |
> | ------- | ---------------- | ---------------- | ---------------- | ---------------- | ---------------- |
> | Entropy | 15.65 $\pm$ 0.08 | 17.96 $\pm$ 0.28 | 18.01 $\pm$ 0.16 | 18.12 $\pm$ 0.17 | 18.91 $\pm$ 0.16 |
>
> The states sampled using SD3 exhibit higher entropy, indicating a better capability to learn diverse behaviors.
>
> [1] Laskin, Michael, et al. "CIC: Contrastive intrinsic control for unsupervised skill discovery."
>
> [2] Yang, Rushuai, et al. "Behavior contrastive learning for unsupervised skill discovery." International Conference on Machine Learning. PMLR, 2023.
>
> [3] Mazzaglia, Pietro, et al. "Choreographer: Learning and adapting skills in imagination." arXiv preprint arXiv:2211.13350 (2022).
>
> **W2**: Comparisons are not fair.
>
> **Response**: Thanks for the comment. To optimize $I_{SD3}$, we need to estimate $d^{\pi}_z(s)$. Given the broad state space and the shared network parameters across different skills, the traditional CVAE approach may result in a loss of estimation accuracy to some extent. To address this, we introduce a soft-modularized CVAE. Additionally, certain MI-based methods neither require nor can benefit from soft modularization. For instance, the core of DIAYN lies in training a discriminator to identify the skill associated with each state, thereby optimizing the reverse form of MI. In this process, there is no need to apply soft modularization to the discriminator.

---

> ### Author Response · Authors · 2024-11-24
> **Response to Reviewer z324(Part 2/3)**
>
> Based on your suggestion, we conducted experiments on 12 downstream tasks without using the soft-modularized CVAE, and the results are shown in the table below. Even without the soft-modularized CVAE, our method still achieves competitive performance on several downstream tasks.
>
>
> | Task              | CIC          | BeCL         | SD3          | SD3(w/o soft_modu) |
> | ----------------- | ------------ | ------------ | ------------ | ------------------ |
> | Quad Stand        | 700 $\pm$ 55 | 875 $\pm$ 33 | 847 $\pm$ 17 | 752 $\pm$ 64       |
> | Quad Walk         | 621 $\pm$ 69 | 743 $\pm$ 68 | 752 $\pm$ 40 | 642 $\pm$ 80       |
> | Quad Run          | 445 $\pm$ 36 | 535 $\pm$ 13 | 471 $\pm$ 13 | 422 $\pm$ 34       |
> | Quad Jump         | 565 $\pm$ 44 | 727 $\pm$ 15 | 676 $\pm$ 29 | 589 $\pm$ 45       |
> | Walker Stand      | 959 $\pm$ 2  | 952 $\pm$ 2  | 930 $\pm$ 5  | 910 $\pm$ 16       |
> | Walker Walk       | 903 $\pm$ 21 | 883 $\pm$ 34 | 914 $\pm$ 11 | 870 $\pm$ 30       |
> | Walker Run        | 450 $\pm$ 19 | 387 $\pm$ 22 | 451 $\pm$ 23 | 409 $\pm$ 55       |
> | Walker Flip       | 641 $\pm$ 26 | 611 $\pm$ 18 | 595 $\pm$ 25 | 523 $\pm$ 33       |
> | Jaco Top Left     | 149 $\pm$ 10 | 125 $\pm$ 10 | 142 $\pm$ 7  | 125 $\pm$ 5        |
> | Jaco Top Right    | 163 $\pm$ 9  | 126 $\pm$ 10 | 152 $\pm$ 7  | 117 $\pm$ 5        |
> | Jaco Bottom Left  | 154 $\pm$ 6  | 148 $\pm$ 13 | 151 $\pm$ 7  | 134 $\pm$ 8        |
> | Jaco Bottom Right | 149 $\pm$ 4  | 139 $\pm$ 14 | 152 $\pm$ 9  | 122 $\pm$ 8        |
>
> **W3**: The method could be sensitive to hyperparameters, e.g. softmax temperature.
>
> **Response**: Thanks for the comment. In SD3, the softmax function is used in the routing network to determine the weight of each module in one layer with respect to the corresponding modules in the next layer. According to research by Geoffrey Hinton et al.[1], when the temperature parameter in the softmax function is set to a relatively large or small value, the output of the softmax gradually converges to either a uniform distribution or a deterministic value. In such cases, the functionality of the routing network is weakened or disrupted, leading to a degradation in the performance of SD3. Based on our findings, we conclude that setting the temperature to its default value of 1 is reasonable.
>
> Additionally, we propose that in any scenario where softmax is used to compute weights, extreme temperature settings can negatively affect network performance. This phenomenon is inherently linked to the principles of the softmax function itself.
>
> The number of modules $m$ is setting to 4.
>
> [1] Geoffrey Hinton, Oriol Vinyals, and Jeff Dean. Distilling the knowledge in a neural network. arXiv preprint arXiv:1503.02531, 2015.
>
> **Q1**: Why use PPO for the maze visualization example, while DDPG is used as the main results in URLB?
>
> **Response**: Thanks for the question.  We follow the experiment setting of state-based URLB and use DDPG as the backbone. This allows us to directly leverage the DDPG hyperparameters provided in URLB, such as the target network update rate. On one hand, this reduces the time cost on tuning backbone hyperparameters, and on the other hand, it ensures a fairer comparison between methods.
>
> For the maze experiments, we adopt an on-policy PPO backbone. This choice is motivated by two considerations: (i) the maze environment is simpler compared to state-based URLB, and (ii) the primary goal of this experiment is to validate the basic effectiveness of the algorithm and provide visualization results. Using on-policy PPO helps to minimize the introduction of additional hyperparameters and enables faster convergence to the desired evaluation outcomes.
>
> **Q2**: Softmax temperature is not mentioned in Line 212.
>
> **Response**:  Thanks for the question. We will correct the typo in the revised version.
>
> The softmax temperature is inherently a hyperparameter of the softmax function itself and is not an exclusive component of our soft-modularized CVAE structure. Therefore, it was not mentioned in Line 212.
>
> **Q3**: The noise is added on states, or transitions (states and actions)?
>
> **Response**: Thanks for the question. The noise is added on states. We added a layer of NoiseWrapper to the environment, which injects noise sampled from $N(0, 0.1)$ into the state.
>
> **Q4**: The results in Table 2 and Table 3 are partially highlighted, which is confusing.
>
> **Response**: Thanks for the question. The highlighted results in Table 2 and Table 3 indicate the two highest mean scores across various downstream tasks. And, we will clarify this in the revised version.

---

> > ### Author Response · Authors · 2024-11-24
> > **Response to Reviewer z324(Part 3/3)**
> >
> > **Q5**: What's the number of modules in cVAE?
> >
> > **Response**: Thanks for the question. The number of modules ($m$) is 4. We conduct ablation experiments on the number of modules in the state-based quadruped environment, and the results are shown in the table below.
> >
> > | Task       | Quad Stand | Quad Walk | Quad Run | Quad Jump |
> > | ---------- | ---------- | --------- | -------- | --------- |
> > | Module = 2 |777 $\pm$ 26|638 $\pm$ 45|377 $\pm$ 38|499 $\pm$ 32|
> > | Module = 3 |862 $\pm$ 25|650 $\pm$ 29|456 $\pm$ 26|590 $\pm$ 20|
> > | Module = 4 |847 $\pm$ 17|752 $\pm$ 40|471 $\pm$ 13|676 $\pm$ 29|
> > | Module = 5 |781 $\pm$ 31|799 $\pm$ 31|390 $\pm$ 32|541 $\pm$ 24|
> > | Module = 6 |680 $\pm$ 27|323 $\pm$ 32|261 $\pm$ 31|375 $\pm$ 34|
> >
> > From the table, it can be observed that the performance differences are minimal when the number of modules is set to 2, 3, or 5. The best performance is achieved when the number of modules is set to 4. However, when the number is increased to 6, there is a significant performance drop. We attribute this to the difficulty in effectively training the soft-modularized structure as the number of modules becomes too large.

---

### Meta-Review · Area_Chair_NJgf · 2024-12-18

**Metareview:**

This paper presents an unsupervised RL method for maximizing state visitation of the skills trained. All reviewers agree that there are interesting aspects in the formulation and value in the analysis of the paper. However, the sticking point is the experiments. Reviewers find the experimental results weak as the results are not competitive to state of the art skill discovery methods. I would hence recommend the authors to find more challenging benchmarks in addition to URLB. This will allow for showcasing larger gains with SD3.

**Additional Comments On Reviewer Discussion:**

Several concerns were raised. The reviewers' comments nudged some review scores up, but not substantially to push it over acceptance.

---

### Decision · Program_Chairs · 2025-01-22

Reject